# saRNA vaccine expressing membrane-anchored RBD elicits broad and durable immunity against SARS-CoV-2 variants of concern

Mai Komori [1,10], Takuto Nogimori [2,10], Amber L. Morey [1], Takashi Sekida[3], Keiko Ishimoto [1], Matthew R. Hassett[1], Yuji Masuta [2], Hirotaka Ode[4], Tomokazu Tamura [5], Rigel Suzuki[5], Jeff Alexander[1], Yasutoshi Kido [6], Kenta Matsuda[1], Takasuke Fukuhara [5], Yasumasa Iwatani [4,7], Takuya Yamamoto [2,8,9] ✉, Jonathan F. Smith [1] ✉ & Wataru Akahata [3] ✉

Several vaccines have been widely used to counteract the global pandemic caused by SARS-CoV-2. However, due to the rapid emergence of SARS-CoV-2 variants of concern (VOCs), further development of vaccines that confer broad and longer-lasting protection against emerging VOCs are needed. Here, we report the immunological characteristics of a self-amplifying RNA (saRNA) vaccine expressing the SARS-CoV-2 Spike (S) receptor binding domain (RBD), which is membrane-anchored by fusing with an N-terminal signal sequence and a C-terminal transmembrane domain (RBD-TM). Immunization with saRNA RBD-TM delivered in lipid nanoparticles (LNP) efficiently induces T-cell and B-cell responses in non-human primates (NHPs). In addition, immunized hamsters and NHPs are protected against SARS-CoV-2 challenge. Importantly, RBD-specific antibodies against VOCs are maintained for at least 12 months in NHPs. These findings suggest that this saRNA platform expressing RBD-TM will be a useful vaccine candidate inducing durable immunity against emerging SARS-CoV-2 strains.

SARS-CoV-2 has spread rapidly and globally with >607 million cases and 6.4 million deaths, causing a significant impact on global health and economies (WHO Coronavirus (COVID-19) Dashboard, https://covid19.who.int/). SARS-CoV-2 continues to spread with the emergence of new antigenic variants, and broadly protective vaccines that are capable of worldwide distribution are an urgent and high priority. There are currently several approved vaccines against coronavirus disease 2019[1] (COVID-19) that have contributed to reducing the impact of the pandemic. However, due to the relatively short duration of the immune responses induced by these vaccines, as well as the continued

[1]VLP Therapeutics, Inc. 704 Quince Orchard Rd. #110, Gaithersburg, MD 20878, USA. [2]Laboratory of Precision Immunology, Center for Intractable Diseases and ImmunoGenomics, National Institutes of Biomedical Innovation, Health, and Nutrition, Osaka 567-0085, Japan. [3]VLP Therapeutics Japan, Inc. 1-16-4 Nishi-Shinbashi, Minato-ku, Tokyo 100-0003, Japan. [4]Clinical Research Center, National Hospital Organization Nagoya Medical Center, Nagoya, Aichi 460-0001, Japan. [5]Department of Microbiology and Immunology, Faculty of Medicine, Hokkaido University, Hokkaido 060-8638, Japan. [6]Department of Virology & Parasitology, Graduate School of Medicine, Osaka Metropolitan University, Osaka 545-0051, Japan. [7]Division of Basic Medicine, Nagoya University Graduate School of Medicine, Nagoya 466-8550, Japan. [8]Laboratory of Aging and Immune Regulation, Graduate School of Pharmaceutical Sciences, Osaka University, Osaka 565-0871, Japan. [9]Department of Virology and Immunology, Graduate School of Medicine, Osaka University, Osaka 565-0871, Japan. [10]These authors contributed equally: Mai Komori, Takuto Nogimori. ✉e-mail: yamamotot2@nibiohn.go.jp; jsmith@vlptherapeutics.com; wakahata@vlptherapeutics.com

emergence of new VOCs[2], further development of vaccine designs is required.

In this study, we utilized a self-amplifying RNA (saRNA) platform technology to develop a COVID-19 vaccine. This single-cycle vector system utilizes an alphavirus RNA amplification system, the Venezuelan Equine Encephalitis Virus (VEEV)-based replicon expression vector[3]. This vector expresses the alphavirus nonstructural proteins (nsPs)1-4, which together replicate and transcribe the saRNA resulting in efficient expression of the gene(s) of interest. Due to this self-amplification process, the level and duration of expression of target antigens is higher and longer than that observed with mRNA vaccine platforms[4]. Therefore, prolonged presentation of the antigen to the immune system using this saRNA platform is expected. There are several preclinical reports using saRNA vectors to develop COVID-19 vaccine candidates[5–10] expressing full length S protein, and these candidate vaccines induced high humoral and cellular responses as well as protection against COVID-19 challenge in animal models.

It has been reported that over 90% of neutralizing antibodies from infected patients target the SARS-CoV-2 S Receptor Binding Domain (RBD)[11]. Given the correlation between neutralizing antibody and protection, we designed the transgene inserted into the saRNA vector based on RBD sequences to focus the immune response on the highest concentration of neutralizing epitopes.

It has been shown that efficient presentation of antigens to B cell receptors in multivalent arrays leads to a stronger signal induction[12]. To induce B cell responses in this manner, we fused the S protein signal sequence to the N terminus of the RBD, and the transmembrane-cytoplasmic tail domain (TM) from influenza hemagglutinin protein (HA) to the C terminus of the RBD[13]. This approach results in the display of the RBD sequence on the surface of transfected cells in a multivalent manner. In this study, we report the development of this RBD-based saRNA candidate vaccine, and the preclinical evaluation in mice, hamster and non-human primate models using lipid nanoparticle delivery systems.

## Results

### Designing an saRNA vaccine expressing a membrane-anchored, SARS-CoV-2 spike RBD

We first designed candidate RBD vaccines and compared the expression characteristics of secreted RBD and membrane-anchored RBD (RBD-TM) constructs. The secreted RBD version was constructed using the SARS-CoV-2 signal sequence (amino acids [aa] 1-13 of S) fused to the Wuhan-Hu-1 RBD sequence (aa 330-521 of S) (Fig. 1a, top). Our initial RBD-TM construct was expressed as a fusion protein with a murine CD80 transmembrane and cytoplasmic tail domain, that resulted in display of the RBD immunogen on the surface of transfected cells. (Fig. 1a, bottom). The expression of secreted and membrane-anchored RBDs at ~30 and ~40 kDa, respectively, was confirmed by Western blotting after transfection of the saRNA vectors into Human Embryonic Kidney 293 T (293 T) cells (Fig. 1b). The secreted RBD was detected in both the supernatant and cell lysate fractions while the expression of RBD-TM was detected only in the cell-associated fraction, indicating the RBD-TM remains as a membrane-anchored protein. To compare the relative immunogenicity of the secreted RBD and RBD-TM constructs, mice were immunized with these saRNA vectors packaged in VEEV particles. As shown in Fig. 1c, the titers of RBD-binding IgG antibody induced by RBD-TM are significantly higher than those induced by the secreted RBD (19.3-fold higher after 2nd injection, and 9.2-fold higher after 3rd injection, $P < 0.01$, respectively). Antibody assays monitoring the inhibition of RBD binding to the angiotensin-converting enzyme 2 (ACE2) also demonstrated that significantly higher titers were induced by RBD-TM compared to the secreted RBD (11.7-fold higher after 3rd injection, $P < 0.01$, Fig. 1d).

To further optimize this immunogen design, we tested an extended RBD region sequence to potentially enhance stability. Four pairs of intramolecular disulfide bonds have been reported near and within the RBD region[14]. Three pairs (336aa/361aa, 379aa/432aa and 391aa/525aa) help to stabilize the beta-sheet structure, and the 480aa/488aa pair connects the loops in the distal end of the receptor binding motif (RBM)[14]. Among these four pairs, three (336/361aa, 379/432aa and 480/488aa) were included within the initial RBD (330-521 aa) construct tested. In order to further optimize the RBD structure, we expanded the RBD sequence to include residues 327-531aa to enable the fourth disulfide bond (391/525aa) (Stabilized RBD) as depicted in Fig. 1e. We also compared TM sequences derived from SARS-CoV-2 S protein and the influenza hemagglutinin (HA) protein[15] instead of the CD80 sequence used in the original construct to avoid any possibility of inducing autoimmune responses against endogenous CD80 homologs. The original RBD sequence (330-521aa) when fused to the TM region from S or HA both showed relatively low expression of RBD on the surface of the transfected cells. However, improved expression on the surface of the cells was observed when the TM regions derived from SARS-CoV-2 S and influenza HA proteins were fused to the stabilized RBD (Supplementary Fig. 1). These expression data correlated with the immunogenicity of these variants when tested in mice. Mice immunized with the original RBD (330-521aa) fused to TM from S or HA showed a lower level of antibody response, whereas the animals immunized with the stabilized RBD fused to either the TM from S or HA showed a higher antibody response as measured by ELISA or by the ACE2 binding inhibition assay (Fig. 1f, g). Although the overall immunogenicity of stabilized RBD vaccine candidates with TM regions derived from S, or HA were comparable, we opted to proceed with the stabilized RBD with the TM from HA, as the primary responses from HA TM were significantly higher than S TM (Fig. 1f, left). In the remainder of the studies described here, the stabilized RBD with the HA TM is referred to as the RBD-TM.

### Robust humoral and cellular immune responses were elicited by saRNA RBD-TM in mice

We next compared the immunogenicity in mice of the saRNA RBD-TM construct formulated in LNP with an S1 subunit vaccine (Wuhan-Hu-1 sequence) formulated in an alum adjuvant, and observed significantly higher IgG antibody titers against either S1 or RBD antigens with 10 μg of saRNA RBD-TM than with 10 μg of S1 protein with alum adjuvant (Supplementary Fig. 2a). This difference remained significant at both time points tested 4 weeks after the prime and 2 weeks after the booster immunization. saRNA RBD-TM also showed high IgG antibody titers against RBDs bearing either K417N or E484K, and RBDs of three VOCs (Alpha [N501Y], Beta [K417N, E484K and N501] and Gamma [K417T, E484K and N501Y]) antigens (Supplementary Fig. 2b). We also tested sera from these immunized mice in ACE2 inhibition assays against variant RBDs, and saRNA RBD-TM (10 μg) showed significantly higher titers than S1 protein (10 μg) with alum adjuvant against all variants tested (17.2-fold against Wuhan-Hu-1, 59.2-fold against alpha, 23.9-fold against Beta and 27.4-fold against Gamma, Supplementary Fig. 2c). We next tested the neutralizing activities in these mice against a SARS-CoV-2 Wuhan D614G (hereafter, referred to as D614G) as well as Gamma viruses. The inhibitory titers achieved were significantly higher with saRNA RBD-TM (10 μg) than S1 protein (10 μg) with alum against D614G (Supplementary Fig. 2d). The ratio of neutralizing antibody against D614G virus and total IgG against Wuhan-Hu-1 RBD were higher with saRNA RBD-TM vaccine than with the S1 subunit vaccine (Supplementary Fig. 2e). We also directly compared the immunogenicity of RBD-TM and whole S (Wuhan-Hu-1) as a prefusion form (S2P) using the same saRNA platform. We immunized mice with either 1 or 10 μg of saRNA expressing RBD-TM or S2P. The magnitude of binding antibodies was greater in RBD-TM immunized mice when compared to the pre-fusion S2P immunogen (Supplementary Fig. 2f), suggesting that the RBD-based design can be an alternative candidate for SARS-CoV-2 vaccine immunogens.

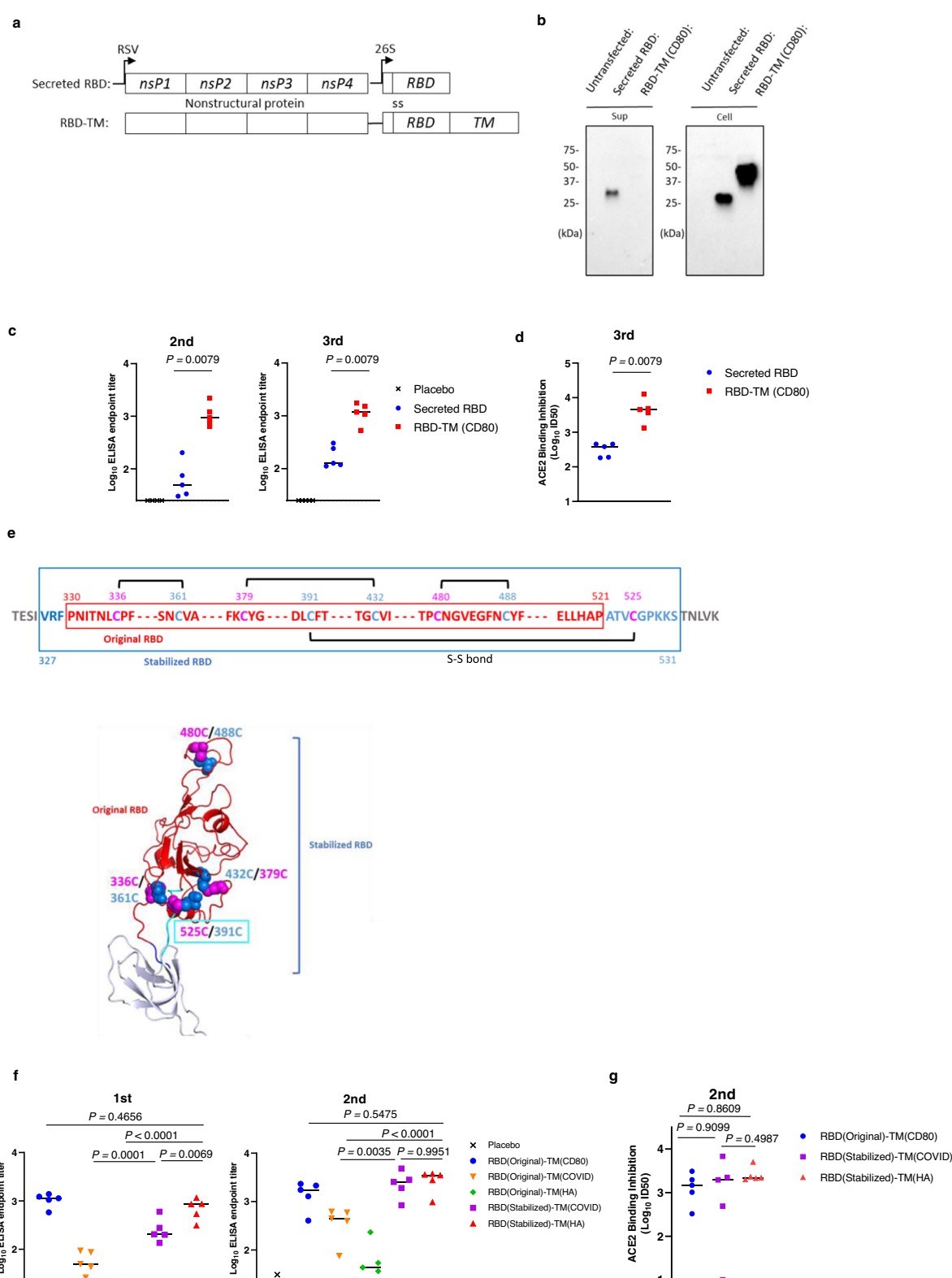

The relative immunogenicity of three SARS-CoV-2 Wuhan-Hu-1 immunogens; 1 μg of saRNA expressing secreted RBD, 1 μg of saRNA expressing RBD-TM, or 10 μg of mRNA expressing full length of S2P antigen (mRNA S2P) were compared in mice. The saRNA and mRNA were transcribed with unmodified nucleotides, and were delivered with the same LNP formulation. After two vaccinations, saRNA expressing RBD-TM induced significantly higher anti-RBD IgG titers against

Wuhan-Hu-1, Gamma, Delta and Omicron (BA.1) compared to saRNA expressing secreted RBD or mRNA expressing the S2P antigen (Fig. 2a). We also measured subclass IgG1 and IgG2a titer in these immunized mice (Fig. 2b, left panel) and the ratio of IgG2a/IgG1 sub-class antibodies was also determined (Fig. 2b, right panel). Among these vaccine groups, the ratio was highest with the RBD-TM vaccine, suggesting that saRNA RBD-TM induced a more Th1-biased immune response.

**Fig. 1 | Vaccine design based on the receptor binding domain and transmembrane (TM) sequences. a** Schematic representation of the saRNA vector used for expressing RBD antigens. The saRNA vector consists of the VEE virus nonstructural polyproteins nsP1, nsP2, nsP3 and nsP4, a signal sequence (ss) and RBD or RBD fused to TM. **b** The Western blot images show RBD proteins detected in the media supernatant (Sup) and 293 T cells (Cell). Each lysate faction was harvested at 48 h post transfection. The experiment was performed at least 3 times and the representative image is shown. **c** BALB/c female mice (*n* = 5 per group) were injected intramuscularly three times with 107 IU/dose of replicon particle expressing SARS-CoV-2 Spike RBD at weekly intervals. Antibody titers of sera from the immunized mice were evaluated by ELISA against SARS-CoV-2 RBD proteins 1 week after the 2nd immunization (Day 14) and 3 weeks after the 3rd (Day 35). Plots represent individual endpoint titers with medians (lines). Mann–Whitney test (two-tailed) was used to determine significance. Experiment was performed two times independently. **d** The ACE2 and RBD binding inhibition titers of sera from the immunized mice (*n* = 5 per group) were measured. Plots indicate the −Log10 ID50 values with medians. Mann–Whitney test (two-tailed) was used to determine significance. Experiment was performed two times independently. **e** Schematic diagram of the antigen is showing the RBD (330-521aa) sequence (red), peptide linkers (327-329 aa) (blue), and (522-531 aa) (cyan). Black lines show the four disulfide bonds (Top panel). Structural models for the stabilized RBD as depicted in ribbon diagrams (Bottom panel) showing the location and structural orientation of the four disulfide bonds. RBD was modeled based on Protein Data Bank (PDB) code 6XM4 and displayed using PyMOL (http://www.pymol.org). Amino acids shown as pink and blue spheres are the cysteine formed disulfide bonds. **f** BALB/c female mice (*n* = 5 per group) were immunized intramuscularly twice at a four-week interval with 2.2 × 106 IU/dose of replicon particle expressing SARS-CoV-2 Spike RBD (original or stabilized) with TM. Antibody titers of sera from the immunized mice were determined using ELISA against SARS-CoV-2 Spike RBD protein. Plot represents individual endpoint titers with medians. One-way ANOVA was used to determine significance. Experiment was performed two times independently. **g** The ACE2 and RBD binding inhibition titers of sera from the immunized mice (*n* = 5 per group) were measured at 3 weeks after 2nd immunization (Day 42). Plots represent the −Log10 ID50 values with medians. One-way ANOVA was used to determine significance. Experiment was performed two times independently.

We next measured the frequency of antigen-specific B and T cells in splenocytes (Supplementary Fig. 3a and 3b). To quantify antigen-specific B cells, an RBD probe against Wuhan-Hu-1 was used[16]. The results showed that the frequency of RBD Wuhan-Hu-1-binding B cells in saRNA RBD-TM vaccinated mice was higher than in mice vaccinated with saRNA expressing secreted RBD or mRNA expressing S2P (4.7-fold and 4-fold, respectively; Fig. 2c, Supplementary Fig. 3c). This suggests that saRNA RBD-TM vaccine induces B cell memory more efficiently, which may contribute to longer B cell antibody production. Moreover, when antigen-specific B cells were measured with three differentially labeled probes against Wuhan-Hu-1, Delta or Omicron (BA.1) RBD, we observed that the RBD-specific B cells in the saRNA RBD-TM vaccinated mice reacted with the probes against all three or two of probes against Wuhan-Hu-1 and Delta (Fig. 2d). This suggests that saRNA RBD-TM vaccinated mice could elicit antibodies which are cross reactive against all three Wuhan-Hu-1, Delta and Omicron (BA.1) RBD, or two of Wuhan-Hu-1 and Delta RBD.

To compare the T cell responses induced by these vaccines, we stimulated splenocytes with Wuhan-Hu-1 peptide pools against RBD or S proteins and measured IFN-γ, TNF and IL-2 cytokine expression from the T cells. A representative flow cytometry analysis from these studies is presented in Supplementary Fig. 3d and 3e. Both saRNA RBD-TM and saRNA secreted RBD induced stronger CD4 Th1 type responses against Wuhan-Hu-1 than the mRNA S2P construct (Fig. 2e) and against Delta and Omicron (BA.1) (Supplementary Fig. 3f). These differences were especially apparent with respect to the production of IFN-γ. These data suggest that antigen-specific CD4+ T cells elicited by both saRNA vaccines are cross-reactive with multiple VOCs. Furthermore, more polyfunctional subpopulations (+IFN-γ + TNF + IL-2) from the specific CD4+ T cells were induced in both saRNA vaccines compared to mRNA S2P vaccine (Fig. 2f).

We also evaluated the correlation between the frequency of Th1 responses against RBD, anti-RBD IgG endpoint titers, and the frequency of RBD-specific B cells. A correlation was observed between the frequency of Th1 in CD4 T cells responses and anti-RBD IgG endpoint titers (Fig. 2g). This correlation suggests that Th1 in CD4 T cell responses are important for the induction of strong antibody responses in saRNA vaccination.

We also measured the frequency of antigen-specific CD8+ T cells against Wuhan-Hu-1, Delta and Omicron (BA. 1) RBD. saRNA vaccines efficiently induced specific CD8+ T cells expressing CD107a, IFN-γ or TNF against Wuhan-Hu-1 RBD (Fig. 2h) as well as against Delta and Omicron (BA. 1) RBD (Supplementary Fig. 3g) compared to mRNA S2P. We also observed that the frequencies of the subpopulations simultaneously expressing CD107a/IFN-γ/TNF and CD107a/IFN-γ were higher from both saRNA vaccines than from mRNA S2P (Fig. 2i).

We measured the blood cell counts in the mice after immunization with saRNA (secreted RBD vs. RBD-TM). The mice immunized with secreted RBD showed higher white blood cells (WBC) and granulocytes (GRA) compared to the mice immunized with RBD-TM (*P* < 0.01), suggesting that the RBD-TM may have lower reactogenicity than secreted RBD (Supplementary Fig. 4).

Taken together, these results indicate that saRNA vaccines, especially saRNA RBD-TM, can elicit potent humoral and cellular immune responses.

## Protective efficacy of saRNA RBD-TM in a NHP challenge model

To assess candidate vaccine immunogenicity and efficacy in NHPs, Cynomolgus monkeys were immunized with two doses of 50 μg of saRNA RBD (Wuhan-Hu-1)-TM delivered in LNP on week 0 and week 4. The saRNA RBD(Wuhan-Hu-1)-TM vaccine induced IgG antibodies against RBD from Wuhan-Hu-1, Beta, Gamma and Delta (Fig. 3a). In addition, the NHPs were challenged with live SARS-CoV-2 Wuhan-Hu-1 at week 8 post immunization. Bronchoalveolar lavage (BAL) and nasal/oropharyngeal swabs were collected at 2- and 4-days post challenge, and the viral titers were measured (Fig. 3b). A significant decrease in viral titers was detected in the BAL of immunized animals at 2 days post infection (*P* < 0.05) and the same trend was observed in nasal/oro-pharyngeal swabs in both timepoints. Viruses were undetectable in the BAL at both 2 and 4 days after challenge. These results suggest that saRNA RBD-TM can confer protection against infection with the homologous SARS CoV-2 virus.

## Induction of cross-reactive and protective antibody responses by saRNA RBD (Gamma)-TM in hamsters

To potentially enhance and broaden the immunogenicity of the saRNA RBD-TM platform, we designed a construct that expressed the homologous RBD sequences derived from the SARS CoV-2 Gamma variant (saRNA RBD (Gamma)-TM). The RBD sequence from Gamma has three amino acid mutations (K417T, E484K, N501Y) compared to Wuhan-Hu-1. The immunogenicity and protective efficacy of this construct was compared to the saRNA RBD(Wuhan-Hu-1)-TM in Golden Syrian hamsters. Hamsters were immunized with two doses of vaccine delivered with LNP at week 0 and 4, followed by intra-nasal challenge with either SARS CoV-2 Wuhan-Hu-1 or the Gamma variant at week 8. We observed that both vaccines induced IgG binding antibodies against Wuhan-Hu-1, Gamma, Delta, and Omicron (BA.1, BA.2 and BA.3) RBD (Fig. 4a). The group immunized with saRNA RBD (Gamma)-TM induced significantly higher IgG antibody titers against Omicron variants compared to the group immunized with saRNA RBD (Wuhan-Hu-1)-TM (*P* < 0.05). Neutralizing activities were also tested against SARS-CoV-2 D614G and three variants (Fig. 4b). saRNA RBD

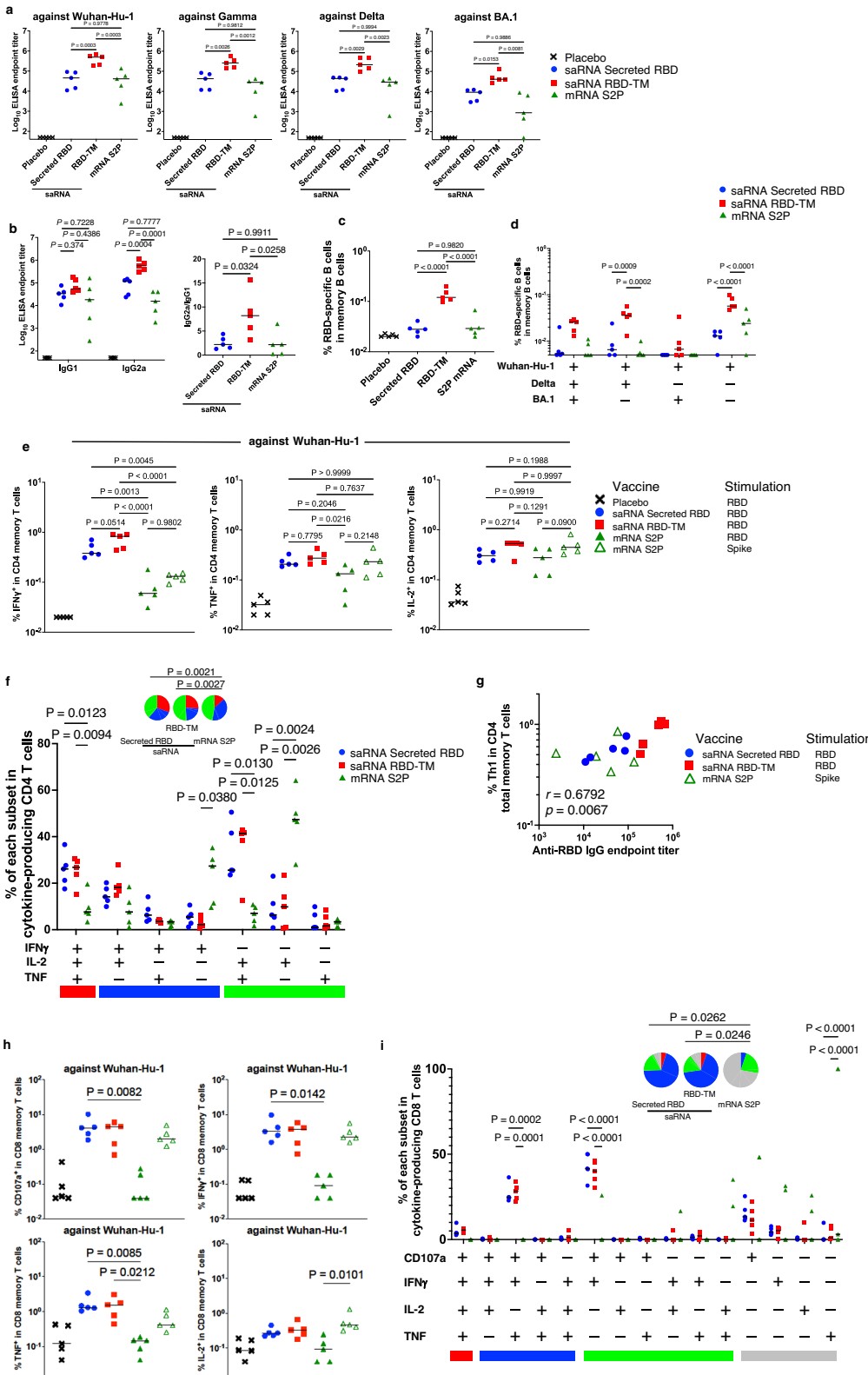

(Gamma)-TM induced significantly higher neutralization titers against Gamma, and Omicron (BA.1) than saRNA RBD Wuhan-Hu-1, but comparable levels against D614G and Omicron (BA.2). To compare the protective efficacy of the saRNA vaccine variants, the body weight of hamsters was monitored for 8 days post Wuhan-Hu-1 or Gamma SARS-CoV-2 virus challenges. In Wuhan-Hu-1 challenged groups, both vaccine groups maintained their weight at the normal level during the

observation period, while significant reduction in weight was observed in placebo group (maximum −15% at Day 7). In Gamma virus challenged groups, the group vaccinated with saRNA RBD(Gamma)-TM prevented weight loss, while moderate weight loss was observed in the group vaccinated with the heterologous saRNA RBD(Wuhan-Hu-1)-TM. A significant weight loss was observed again in the placebo group (Fig. 4c).

**Fig. 2 | Comparison of the immunogenicity of the RBD vaccine and a spike mRNA vaccine in mice.** BALB/c female mice ($n = 5$ per group) were immunized intramuscularly twice with a 4 week interval with 1 μg of saRNA-LNP expressing SARS-CoV-2 Spike variant RBD with or without TM or with 10 μg of mRNA-LNP expressing the Spike protein (S2P). All experiments were performed once. **a**, **b** Antibody titers of sera (at Day 42) from the immunized mice were evaluated by ELISA against SARS-CoV-2 RBD from Wuhan-Hu-1, Gamma, Delta, and Omicron BA.1 variants. Plots represent individual endpoint titers with medians (lines). Ratio of IgG2a to IgG1 was calculated based on the endpoint titers. One-way ANOVA was used to determine significance. **c** Frequencies of RBD-specific B cells in memory B cells (gated on CD3⁻NK1.1⁻CD19⁺B220⁺CD138⁻CD38⁺) were measured by flow cytometric analysis. One-way ANOVA was used to determine significance. **d** Frequencies of RBD VoCs cross-binding B cells as percentages of memory B cells. Two-way ANOVA was used to determine significance. **e** Dot plots represent the frequencies of IFN-γ, TNF, or IL-2-secreting CD4⁺ T cells responding to SARS-CoV-2 spike variant RBD peptides in CD4⁺ total memory cells. The lines show the medians. One-way ANOVA was used to determine significance. **f** Frequencies of RBD-specific CD4⁺ T cell subpopulations producing IFN-γ, TNF, and/or IL-2 in cytokine secreting CD4⁺ total memory cells. The x-axis displays each response pattern, whose composition is denoted with a plus for the presence of IFN-γ, TNF and IL-2. The response patterns are grouped and color-coded by number of positive functions and summarized in pie chart form where each pie slice represents the mean proportion of the total CD4⁺ T cell response contributed by response patterns that have all three

(red), any combination of two (blue) or one (green) of the measured functions from each group immunized with saRNA-LNP expressing SARS-CoV-2 Spike variant RBD with (RBD-TM) or without (Secreted RBD) HA TM or 10 μg of mRNA-LNP expressing Spike protein (mRNA S2P). Two-way ANOVA and permutation test were used to determine significance. **g** Dot plots represents the correlation between anti-RBD endpoint titer and frequency of Th1 (producing IFN-γ, TNF or IL-2 responding to RBD peptides). Correlations and two-sided p-values were calculated using the nonparametric Spearman's rank test. **h** Dot plots represent the frequencies of CD107a, IFN-γ, TNF, or IL-2-secreting CD8⁺ T cells responding to SARS-CoV-2 spike variant RBD peptides in CD8⁺ total memory cells. The lines show the medians. One-way ANOVA was used to determine significance. **i** Frequencies of RBD-specific CD8⁺ T cell subpopulations producing CD107a, IFN-γ, TNF, and/or IL-2 in cytokine secreting CD8⁺ total memory cells. The x-axis displays each response pattern, whose composition is denoted with a plus for the presence of CD107a, IFN-γ, TNF and IL-2. The response patterns are grouped and color-coded by number of positive functions and summarized in pie chart form where each pie slice represents the mean proportion of the total CD8⁺ T cell response contributed by response patterns that have all four (red), any combination of three (blue), two (green) or one (gray) of the measured functions from each group immunized with saRNA-LNP expressing SARS-CoV-2 Spike variant RBD with (RBD-TM) or without (Secreted RBD) HA TM or 10 μg of mRNA-LNP expressing Spike protein (mRNA S2P). Two-way ANOVA and permutation test were used to determine significance.

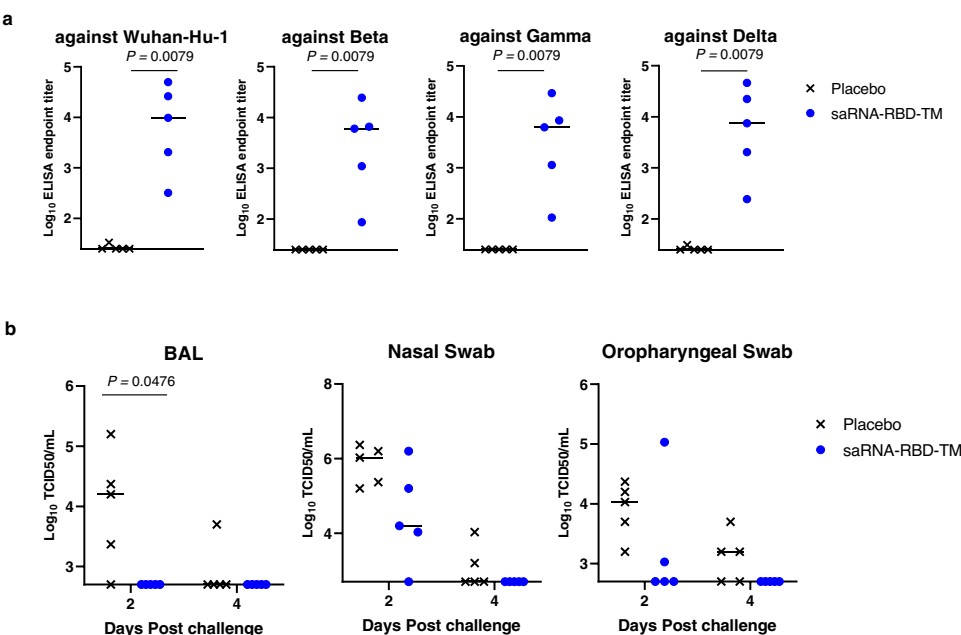

**Fig. 3 | Vaccine efficacy using a non-human primate challenge model.** Adult cynomolgus macaques ($n = 5$ per group) were injected intramuscularly twice with 50 μg of saRNA-LNP expressing SARS-CoV-2 Spike RBD with HA TM at 0 and 4 weeks. The immunized monkeys were challenged with Wuhan-Hu-1 SARS-CoV-2 via the combination intranasal/intratracheal (IN/IT) route on Day 56. **a** Antibody titers of sera (Day 54) from the immunized monkeys were evaluated by ELISA

against SARS-CoV-2 variant Spike RBD proteins. Plots represent individual endpoint titers with medians. Mann–Whitney U-test was used to determine significance. Experiment was performed two times independently. **b** Infectious virus titers of BAL (left panel), nasal swab (center panel) and oropharyngeal swab (right panel) samples collected at days 2 and 4 post challenge were analyzed by TCID₅₀ assay.

Taken together, the data indicate that homologous RBD sequences can be successfully exchanged in this vaccine design, and the insertion of the RBD sequence from the Gamma variant enhanced the cross-reactive immunity against SARS CoV-2 variants and resulted in improved protection against variant SARS CoV-2 infections.

## Duration of immune responses induced by saRNA RBD-TM vaccination in the non-human primate

Cynomolgus macaques were immunized intramuscularly with 45 μg of saRNA RBD(Wuhan-Hu-1)-TM or saRNA RBD (Gamma)-TM delivered in LNP at weeks 0 and 4 to compare the duration of immune response

(Fig. 5a). The anti-Wuhan-Hu-1 RBD (Fig. 5b left panel) and anti-Gamma RBD (Fig. 5b right panel) specific IgG levels in plasma were measured. The standard WHO sera was included in the panel both as an internal control and as a standardized control to allow direct comparison of our results with the datasets from other groups. The immunization with saRNA RBD (Gamma)-TM induced higher RBD-specific IgG titers than saRNA RBD(Wuhan-Hu-1)-TM against Wuhan-Hu-1 (6.6-fold at 6 weeks, 2.2-fold at 28 weeks) and Gamma (82.4-fold at 6 weeks, 8.0-fold at 28 weeks). To assess the durability of the antibody responses, we monitored the antibody titer for 52 weeks. Next, we have calculated the estimated half-life of binding antibody response in NHPs immunized

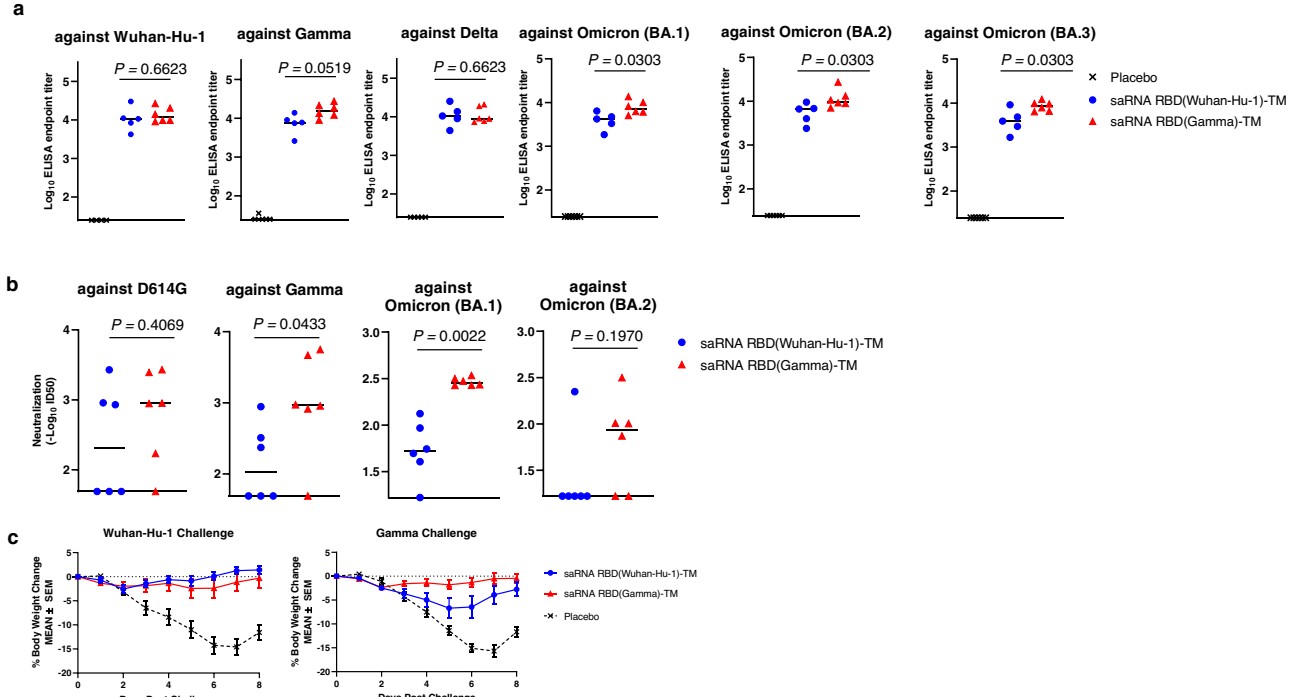

**Fig. 4 | Immunogenicity and cross reactivity of RBD-Gamma based vaccine in hamsters.** Golden Syrian hamsters ($n = 6$ per group) were injected intramuscularly twice with 10 μg of saRNA-LNP expressing SARS-CoV-2 Spike RBD Wuhan-Hu-1 or Gamma at weeks 0 and 4. Hamsters were challenged with SARS-CoV-2 Wuhan-Hu-1 and Gamma at Day 56. **a** Antibody titers of sera (at Day 55) from the immunized hamsters were evaluated by ELISA against SARS-CoV-2 RBD antigens from Wuhan-Hu-1, Gamma, Delta and Omicron (BA.1, BA.2, and BA.3) variants. Plots represent individual endpoint titers with medians. Mann–Whitney test was used to determine significance. Experiment was performed two times independently. **b** SARS-CoV-2 Neutralization assays against the D614G, Gamma, and Omicron (BA.1 and BA.2) were performed to determine the level of neutralizing antibodies in these immunized mice sera at Day 42. Plots indicate the $-\text{Log}_{10}\text{ID}_{50}$ with medians. Experiment was performed two times independently. **c** Body weight changes are shown as mean ± SEM for 8 days after challenges.

with saRNA RBD(Wuhan-Hu-1)-TM and saRNA RBD(Gamma)-TM between months 3 and 7 against Wuhan-Hu-1 and Gamma antigens. The estimated half-life of binding antibody response induced by saRNA RBD(Wuhan-Hu-1)-TM was 178.5 days against Wuhan-Hu-1 and 149.8 days against Gamma, whereas the half-life of binding antibody response induced by saRNA RBD(Gamma)-TM was 198.8 days against Wuhan-Hu-1 and 112.7 days against Gamma, calculated with the use of power-law model. Importantly, the animals immunized with saRNA RBD(Gamma)-TM maintained significantly higher IgG antibody titers against both Wuhan-Hu-1 and Gamma compared to the standard WHO sera throughout the 52-week course of study (Fig. 5b). The breadth of antibody responses was also measured. The saRNA RBD(Gamma)-TM induced cross-reactive antibodies against all strains tested (Wuhan-Hu-1, Delta, BA.1, BA.2, BA.4 and BA.5) and antibody titers induced by saRNA RBD(Gamma)-TM were higher than those induced by saRNA RBD (Wuhan-Hu-1)-TM against all variants (Fig. 5c).

Neutralization activity was also assessed to confirm that it is maintained throughout the observation period (52 weeks). The neutralization activity of the serum was tested against the pseudotyped virus expressing the S protein homologous to immunogen (Fig. 5d). As expected, we detected significant differences in neutralizing antibody titer between saRNA RBD (Wuhan-Hu-1)-TM and saRNA RBD (Gamma)-TM at weeks 28 (p = 0.05) and 52 (p = 0.05). The neutralizing antibody titer of saRNA RBD (Gamma)-TM animals at 52 weeks was comparable level with that of 28 weeks time point, indicating durable neutralizing antibody response was induced. We also assessed the serum neutralizing activity against the live Omicron variant (BA.5). However, the neutralization activity against BA.5 was detected in only one animal at 52 wks post immunization (Supplementary Fig. 5a).

An additional group of monkeys was immunized with saRNA RBD(Gamma)-TM to assess the effect of dose difference between 45 μg and 15 μg. Although the peak IgG antibody titer with the 45 μg dose was higher than with the lower dose, the antibody titers were equivalent level at 28 weeks (Supplementary Fig. 5b), suggesting that saRNA vaccine dose can be optimized in lower dose.

Next, we performed flow cytometric analysis to assess T/B-cell mediated immune responses. The saRNA vaccines increased the frequencies of germinal center (GC) B cells (PNA⁺IgG⁺) (Supplementary Fig. 5c, d) and Tfh cells (Supplementary Fig. 5e, f) in lymph nodes when measured 2 weeks post-boost. Furthermore, RBD-specific B cells were induced in GC B cells (Fig. 5e left panel and Supplementary Fig. 5g) and memory B cells (Fig. 5e right panel and Supplementary Fig. 5h), especially in the saRNA RBD(Gamma)-TM immunized monkeys.

To assess antigen-specific T-cell responses in lymph nodes, the cells from monkey lymph nodes were stimulated with corresponding RBD peptide pools (Supplementary Fig. 5i–r). Here, antigen-specific CD4 T cells were defined as CD4⁺ total memory T cells expressing CD154, and significant frequencies of antigen-specific CD4 T cells were induced (Fig. 5f left panel), and there was significant induction of Th1 phenotype CD4 memory T cells (Fig. 5f middle and right panel). We observed that saRNA vaccine activated Tfh cells robustly (Fig. 5g), and the phenotype of these antigen-specific CD4 T cells were Th1 biased (Fig. 5h). We found that the frequencies of CD154⁺ CD4⁺ total memory were significantly correlated with the frequency of RBD-binding CD27⁺IgG⁺ B cells in lymph nodes (Fig. 5i, P = 0.043).

Finally, we analyzed antigen-specific B cells in PBMCs longitudinally by flow cytometry. The frequency of RBD-specific B cells in PBMCs from the group vaccinated with saRNA RBD (Gamma)-TM were significantly higher at weeks 10, 18 and 26 than those of groups vaccinated with saRNA RBD(Wuhan-Hu-1)-TM (Fig. 5j). Together, these results suggest that saRNA RBD-TM vaccine, in particular the saRNA RBD(Gamma)-TM candidate, can induce broad and durable antibody response.

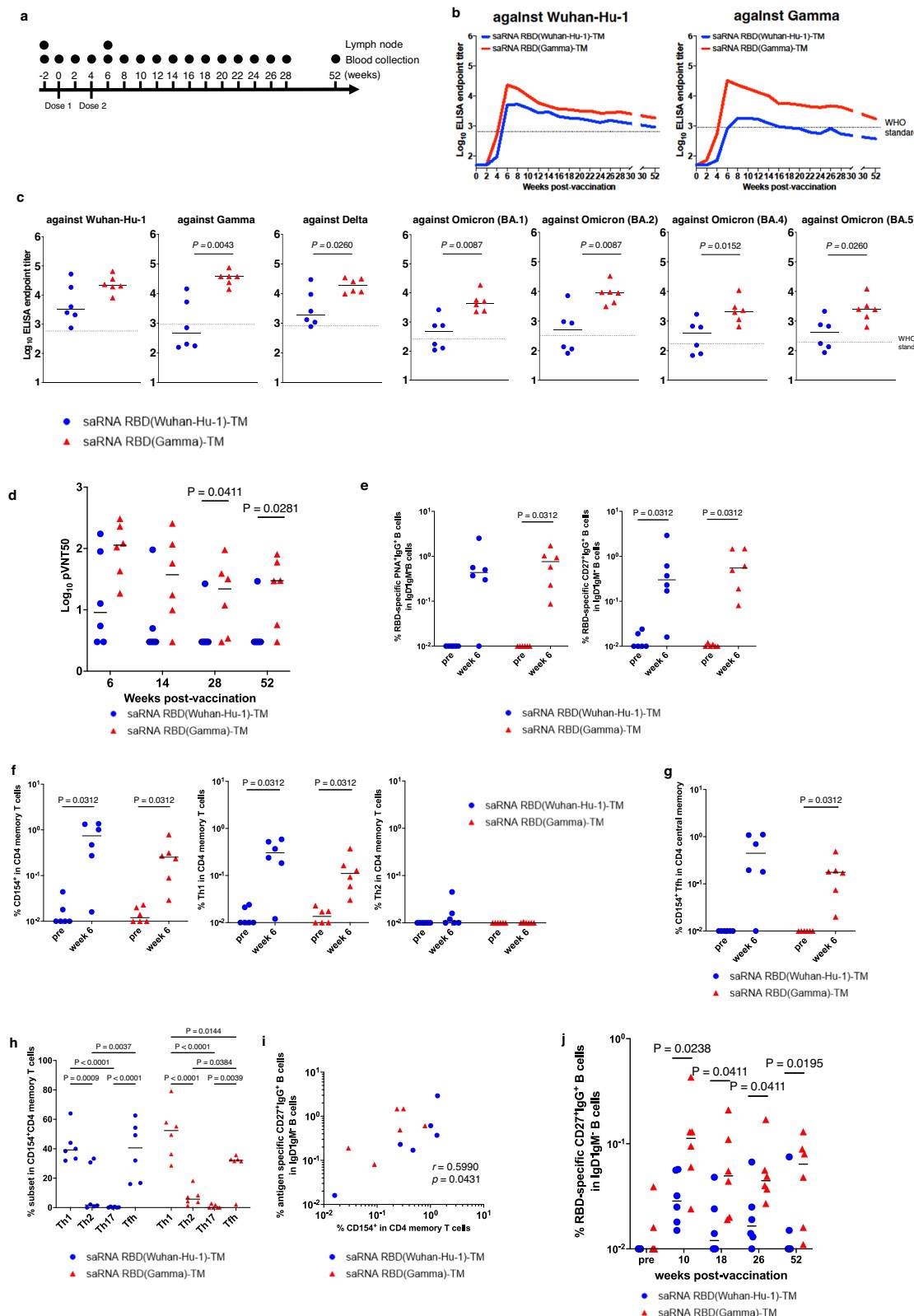

## Discussion

The basic design of the COVID-19 vaccine being tested in these studies is an saRNA vector expressing an optimized RBD sequence fused with a signal sequence at the N-terminus and a membrane anchor at the C-terminus. This design focuses the immune response on the RBD that contains most of the neutralizing epitopes[11], and minimizes non-neutralizing antibodies to other domains that may contribute to

systemic inflammation through antibody-mediated SARS-CoV-2 update by monocytes and macrophages[17] or other mechanisms. Addition of the signal sequence and the membrane anchor resulted in the RBD being efficiently displayed on the surface of transfected cells in a vaccine recipient in a form that we have shown is highly immunogenic in mice, hamsters, and NHP, and efficacious in hamster and NHP challenge models. A concern with the current mRNA vaccines in

**Fig. 5 | Immunogenicity of RBD vaccine in a non-human primate model.**
Cynomolgus macaques (*n* = 6 per group) were immunized intramuscularly twice at a 4 week interval with 45 μg of saRNA-LNP expressing either the SARS-CoV-2 Wuhan-Hu-1 or Gamma RBD with HA TM at a 4-week interval. All experiments were performed once. **a** Schematic of the vaccine study. **b** Antibody titers of plasma from the immunized macaques were evaluated by ELISA against both Wuhan-Hu-1 and Gamma RBD antigens. Dashed lines represent the endpoint titers of the WHO international Standard for anti-SARS-CoV-2 immunoglobulin, human (WHO standard, NIBSC code: 20/136). The standard WHO sera is 1000 binding antibody units (BAU)/mL. **c** Antibody titers of plasma (at week 6) from the immunized macaques were also evaluated by ELISA against SARS-CoV-2 RBD antigens from Wuhan-Hu-1, Gamma, Delta, Omicron BA.1, BA.2, BA.4 and BA.5 variants. Plots represent individual endpoint titers with geometric means (lines). Dashed lines show the endpoint titers of WHO standard. *P*-values (two-sided) were determined using the nonparametric Mann−Whitney *U*-test. **d** Plasma neutralizing antibody titers against pseudo typed virus (pVNT50) in immunized monkeys were measured at weeks 6, 14, 28, and 52. Neutralization was tested against SARS CoV-2 pseudo typed virus bearing the autologous spike protein (Wuhan-Hu-1 spike protein indicated in blue, or Gamma spike protein indicated in red) to the immunogen (saRNA RBD(Wuhan-Hu-1)-TM and saRNA RBD(Gamma)-TM respectively). *P*-values were determined using the nonparametric Mann−Whitney *U*-test. **e** Frequencies of RBD-binding PNA⁺IgG⁺ (left

panel) or CD27⁺IgG⁺ (right panel) B cells as a percentage of IgD⁻IgM⁻ B cells in a lymph node. *P*-values (two-sided) were calculated using the Wilcoxon matched-pairs signed rank test. **f** Dot plots representing the frequencies of CD154⁺ (left panel), Th1 producing IFN- γ, TNF, and/or IL-2 (center panel) and Th2 producing IL-4 and/or IL-13 (right panel) responding to SARS-CoV-2 spike variant RBD peptides in CD4⁺ total memory cells in a lymph node. The lines show the geometric means. *P*-values (two-sided) were calculated using the Wilcoxon matched-pairs signed rank test. **g** Dot plots representing the frequencies of CD154⁺ Tfh responding to SARS-CoV-2 spike variant RBD peptides in CD4⁺ central memory cells in a lymph node. The lines show the geometric means. *P*-values (two-sided) were calculated using the Wilcoxon matched-pairs signed rank test. **h** Frequencies of RBD-specific CD4⁺ T-cell subpopulations Th1 (expressing CXCR3), Th2 (expressing CCR4) Th17 (expressing CCR6) and Tfh (expressing PD-1 and CXCR5) in CD154⁺CD4⁺ total memory cells in a lymph node. Two-way ANOVA was used to determine significance. **i** Correlation between frequency of CD154⁺CD4⁺ total memory cells and frequency of RBD-binding CD27⁺IgG⁺ B cells in a lymph node. Correlations and two-sided *p*-value were calculated using the nonparametric Spearman's rank test. **j** Duration of RBD-binding CD27⁺IgG⁺ B cells in peripheral blood mononuclear cells as percent of IgD⁻IgM⁻ B cells. *P*-values (two-sided) were determined using the nonparametric Mann−Whitney *U*-test.

widespread use is the relatively short duration of the immune responses induced, which is further complicated by the emergence of additional VOCs[2]. Presumably due to the self-amplifying nature of the saRNA vectors, we and others[4] have observed extended in vivo expression of reporter genes with these vectors up to day 86 in some animals (Supplementary Fig. 6), whereas expression from mRNA vectors is diminished within 2 weeks[18,19]. The extended expression may contribute to extended duration of immune responses, although as has been suggested by others, a longer interval between prime and boost vaccinations may be useful to take full advantage of this characteristic. In our NHP study, we demonstrated saRNA RBD(Gamma)-TM induces not only high neutralizing antibody titers, but also stimulates responses in antigen-specific B cells and antigen-specific T cells. Indeed, antibodies induced by saRNA RBD(Gamma)-TM were rarely reduced between 3- and 6-months post-vaccination. Notably, the saRNA RBD(Gamma)-TM has the ability to maintain antigen-specific B cells and plasma neutralizing activity in NHPs throughout the observation period. We speculate that saRNA vaccine could potentially induce high neutralizing antibodies and Th1-biased immune responses in humans by promoting and prolonging the germinal center reaction after vaccination.

We have also shown that this saRNA-RBD vaccine design readily allows for the exchange of homologous RBD sequences from other strains of SARS-CoV-2. The saRNA vaccine expressing Gamma RBD exhibited greater cross-reactive responses against heterologous variants compared to the saRNA expressing Wuhan-Hu-1 RBD. The position of aa 484 is known to have evolved as a mechanism for major immune escape[20], and most of the recent VOC strains contain a mutation at this position, which may contribute to this difference.

The further development of this saRNA vaccine will require evaluation in human clinical trials, and these are ongoing as both a prime and as a booster vaccine. Several other clinical trials using saRNA technology to develop COVID-19 vaccines have also been reported[21,22]. We believe that this saRNA approach will prove useful not only as a vaccine development strategy against newly emerging SARS-CoV-2 variants, but also as a platform for other pandemic pathogenic viruses.

## Methods
### Plasmid construction
The saRNA plasmids encoding the nonstructural protein (nsP) 1, 2, 3 and 4 and the SARS-CoV-2 Spike RBD were constructed based on the Venezuelan Equine Encephalitis Virus (VEEV) Replicon expression vector derived from the TC-83 strain. Briefly, a DNA fragment encoding

the signal sequence, the RBD, and the transmembrane-cytoplasmic domain (TM) were synthesized by Integrated DNA technology (IDT, Coralview, Iowa). The wildtype and stabilized (extended amino acid) RBD sequences were derived from the S gene of SARS-CoV-2 Wuhan-Hu-1 (Genbank# YP_009724390.1) or Gamma (GISAID# EPI_ISL_811149) strain with downstream TM regions derived from the murine B-lymphocyte activation antigen CD80 (aa positions 247-306, Genbank#Q00609.1), Influenza hemagglutinin (HA) (aa positions 518-565, Genbank#P03452.2), or the SARS-CoV-2 S (aa positions 1,204-1,273, Genbank#YP_009724390.1). The DNA fragments were digested with AscI and SbfI enzymes (New England Biolabs), and cloned into the VEEV replicon vector[23] downstream from the subgenomic promotor using the Gibson assembly method[24]. The sequences of both the insert and the boundary regions of the expression plasmids were verified by conventional Sanger sequencing methods.

### Cells
Human embryonic kidney (HEK293T) cells (American Type Culture Collection [ATCC]), Vero kidney epithelial cells (ATCC), VeroE6 (ATCC), and VeroE6 cells constitutively expressing human transmembrane serine protease 2 (TMPRSS2), VeroE6/TMPRSS2 cells (Japanese Collection of Research Bioresources Cell Bank)[25], were maintained in Dulbecco's modified Eagle medium (DMEM) (Thermo Fisher Scientific or Sigma-Aldrich) supplemented with 10% fetal bovine serum (FBS) and 1% Penicillin and Streptomycin (Thermo Fisher Scientific) at 37 °C in a 5% CO₂ incubator. Expi293 cells were purchased from Thermo Fisher Scientific and maintained in Expi293 Expression Medium (Thermo Fisher Scientific).

### Viruses
The SARS-CoV-2 D614G variant JP-NGY003/2020, was isolated in VeroE6 cells from a nasal swab sample of a patient in the Nagoya Medical Center, Japan. SARS-CoV-2, Gamma variant (P.1, TY7-503, GISAID# EPI_ISL_877769), Delta variant (B.1.617.2, TY11-927, GISAID# EPI_ISL_2158617), and Omicron variants (BA1, TY38-873, EPI_ISL_7571617; BA1.1, TY38-871, GISAID# EPI_ISL_7571618; BA2, TY40-385, GISAID# EPI_ISL_9595859) were obtained from the National Institute of Infectious Diseases, Japan. These viruses were amplified in VeroE6 cells, and sequences of the viral full genomes were confirmed by the next-generation sequencing, Illumina MiSeq, with the SARS-CoV-2 genome library preparation reagents, a QIAseq SARS-CoV-2 primer panel (Qiagen), and a QIAseq FX DNA Library Kit (Qiagen). The JP-NGY003/2020 (EPI_ISL_568558) and the TY7-501 (EPI_ISL_833366)

have the D614G mutation and L18F/T20N/P26S/D138Y/G181V/R190S/K417T/E484K/N501Y/D614G/H655Y/T1027I/V1176F mutations in the Spike protein, respectively. The SARS-CoV-2 RNA copy numbers of virus stock were measured by the RT-droplet digital PCR (RT-ddPCR) system (Bio Rad) using a 1-Step RT-ddPCR Advanced Kit for probes (Bio Rad) and a primers (NIID_2019-nCoV_N_F2: AAATTTTGGGGACCAGGAAC and NIID_2019-nCoV_N_R2: TGGCAGCTGTGTAGGTCAAC) and probe (NIID_2019-nCOV_N_P2: 5'-FAM-ATGTCGCGCATTGGCATGGA-TAMRA-3').

set for N2[26]. The end-point titer of each virus stock was determined using VeroE6/TMPRSS2 cells in a 96-well plate format and calculated as the 50% tissue culture infectious dose ($TCID_{50}$)[27] based on 8 replicates for titration. Experiments using SARS-CoV-2 were performed in a biosafety level 3 laboratory.

## Western blotting

Transfected HEK293T cells were lysed in cell lysis buffer (Cell Signaling Technology) and fractionated by SDS-PAGE (Any kD acrylamide gel, Bio-Rad). Proteins were detected by western blotting using anti-SARS-CoV-2 (2019-nCoV) Spike RBD, rabbit polyclonal antibody (1:2000 dilution; Sino Biological) and horseradish peroxidase-conjugated mouse anti-rabbit IgG (1:4000; Santa Cruz Biotechnology). The protein bands were visualized by enhanced chemiluminescence using ChemiDoc™ XRS⁺ (Bio-Rad) and an Image Lab™ Software (Bio-Rad).

## Generation of saRNA encapsulated VLPs

HEK293T cells (~80% confluent) in T-850 flasks (Avantor) with 50 mL of culture media were co-transfected with a plasmid for expression of the VEEV structural proteins (capsid and envelope proteins), and a second plasmid for the transcription of the saRNA SARS-CoV-2 Spike RBD using Fectopro DNA transfection reagent (Polyplus). The culture media were changed 24 h after transfection with FBS-free DMEM, and culture supernatants were harvested at 72 h after transfection and filtered through a 0.45-μm filter (Avantor). Next, the VEEV replicon particles were purified from the filtered culture supernatant using Cellufine sulfate resins (JNC Corporation). Briefly, the Cellufine sulfate resins were washed with water and 300 mM sodium chloride in 10 mM sodium phosphate buffer, and then mixed with culture supernatant (adjusted to 300 mM with 5 M sodium chloride), followed by incubation at room temperature with orbital shaking for 1 h for particle binding. The resins were spun down at 1699 × $g$ for 2 min, and subsequently washed once with 300 mM sodium chloride in 10 mM sodium phosphate buffer. VEEV replicon particles were eluted with 450 mM sodium chloride in 10 mM sodium phosphate buffer, and concentrated with Amicon Ultra-15 centrifugal filter unit-100K (MilliporeSigma). For titration, Vero cells were infected with the concentrated replicon particles and infectious unit (IU) per mL were calculated from the percentage of VEEV-positive cells by Flow Cytometry.

## RBD probe preparation

RBD probes for detection of RBD-specific B cells were obtained by transfection of Expi293 cells with plasmids expressing each RBD variant using the 293fectin transfection reagent (Thermo Fisher Scientific). At day 5 after transfection, the cell supernatants were clarified by centrifugation and filtered. The RBD probes were purified from the supernatant by HisTrap FF column (Cytiva) and HiLoad 16/600 Superdex 200 column (Cytiva) using the AKTA go (Cytiva). The purified RBD probes were biotinylated using a biotin-protein ligase kit (Avidity, Aurora, CO, USA). For flow cytometric analysis, the biotinylated RBD probes were used[28].

## Flow cytometric analysis

For the analysis of cell surface proteins, transfected HEK293T cells were collected and washed with PBS, and stained with an anti-SARS-CoV-2 RBD neutralizing monoclonal antibody (AM128) (1:200 dilution;

ACRO Biosystems) and goat anti-human IgG Fc secondary PE (1:33.3 dilution; eBioscience). The levels of surface proteins were evaluated using an Attune acoustic focusing cytometer (applied biosystems). For analyzing T cells of mice, we performed surface and intracellular cytokine staining of CD4⁺ and CD8⁺ T cells. Briefly, splenocytes from mice were incubated for 30 min in 200 μl RPMI medium containing 10% FBS with or without peptides (17-mers overlapping by 10 residues) corresponding to RBD region of SARS-CoV-2 spike, at a final concentration of 2 μg/ml of each peptide in the presence of anti-CD107a (1D4B, 1:200 dilution). Thereafter, 0.2 μl BD GolgiPlug and 0.14 μl BD GolgiStop (both from BD Biosciences) were added to the cells and incubated for 5.5 h. The cells were then stained using the LIVE/DEAD™ Fixable Aqua Dead Cell Stain Kit (Thermo Fisher Scientific), and stained with anti-CD3 (17A2, 1:200 dilution), anti-CD4 (RM4-5, 1:400 dilution), anti-CD8 (53-6.7, 1:400 dilution), anti-CD62L (MEL-14, 1:200 dilution), anti-CD44 (IM7, 1:2000 dilution), anti-CXCR5 (2G8, 1:40 dilution) and anti-CXCR3 (CXCR3-173, 1:40 dilution) antibodies. After fixation and permeabilization using the Cytofix/Cytoperm kit (BD Biosciences), the cells were stained with anti-IFN-γ (XMG1.2, 1:400 dilution), anti-TNF (MP6-XT22, 1:100 dilution), anti-CD154 (MR1, 1:50 dilution), anti-IL-13 (ebio13A, 1:100 dilution), anti-IL-21 (FFA21, 1:40 dilution), anti-IL-4 (11B11, 1:40 dilution) and anti-IL-2 (JES6-5H4, 1:200 dilution) antibodies. The cells were analyzed using a BD FACSymphony A5 flow cytometer (BD Biosciences). For analyzing of RBD-specific B cells of mice, splenocytes were stained with the LIVE/DEAD Fixable Blue Dead Cell Stain Kit (Thermo Fisher Scientific) for 5 min. Thereafter, the cells were surface-stained with anti-IgM (R6-60.2, 1:67 dilution), anti-B220 (RA3-6B2, 1:2857 dilution), anti-IgD (11-26 c.2a, 1:1000 dilution), anti-CD138 (281-2, 1:100 dilution), anti-NK1.1 (PK136, 1:100 dilution), anti-CD3 (17A2, 1:100 dilution), anti-CD38 (90/CD38, 1:333 dilution) and anti-CD19 (1D3, 1:2000 dilution) antibodies for 15 min. Following a wash step, the cells were incubated with a fluorescent dye-conjugated streptavidin-bound RBD probe for 15 min and analyzed using a BD FACSymphony A5 flow cytometer (BD Biosciences). For analyzing antigen-specific T cells of monkeys, we used the cell staining protocol previously described[29]. For analyzing antigen-specific B cells of monkeys, peripheral blood mononuclear cells and lymph node cells were stained with the LIVE/DEAD Fixable Blue Dead Cell Stain Kit (Thermo Fisher Scientific) for 5 min. Thereafter, the cells were surface-stained with anti-IgD (goat polyclonal antibody, 1:1000 dilution), anti-CD19 (J3.119, 1:10 dilution), anti-CD27 (1A4CD27, 1:25 dilution), anti-CD20 (2H7, 1:200 dilution), anti-IgM (G20-127, 1:100 dilution), anti-IgG (G18-145, 1:200 dilution) and anti-CD3 (SP34-2, 1:100 dilution) antibodies for 15 min. Following wash step, the cells were incubated with fluorescent dye-bound RBD probe for 15 min. The cells were analyzed by a BD FACSymphony A5 flow cytometer (BD Biosciences) using FlowJo Software version 10.8.1 (BD, Ashland, OR) and Simplified Presentation of Incredibly Complex Evaluations version 6.1 provided by Dr. Mario Roederer (National Institutes of Health, Bethesda, MD).

## saRNA synthesis and lipid nano particle (LNP) formulation

Plasmids with RBD variant sequences placed downstream from a T7 promoter were linearized by digestion with the NruI or BspQ1 restriction enzymes at 37 °C or 50 °C for 3 h. The linearized plasmid was then purified using the Wizard Plus SV Miniprep DNA Purification System (Promega), and saRNA was transcribed in vitro using the T7 RiboMAX Express Large-Scale RNA Production System (Promega). After DNase treatment, the saRNA was purified with RNeasy midi kit (Qiagen), and subsequently modified by the addition of a 7-methylguanosine cap with the Vaccinia Capping System (New England Biolabs [NEB]) using the NEB Capping protocol (NEB, M20280). After purification of the capped saRNA using the Monarch kit (NEB), LNP-saRNA was formulated with GenVoy-ILM (Precision Nano Systems) with the NanoAssemblr Ignite (Precision Nano Systems). For the NHP immunogenicity and efficacy studies, the saRNA was

encapsulated in LNPs using a microfluidics method in which an aqueous solution of saRNA at pH=4.0 is rapidly mixed with an ethanol solution of lipids from the FUJIFILM Corporation. LNPs used in this study contained a propriety mixture of ionizable lipid. phospholipid, cholesterol, and PEG-lipid. The proprietary lipids and LNPs are prepared with reference to the patent (WO2021/095876) and supplied by FUJIFILM Corporation. They had mean hydrodynamic diameters of 120-130 nm as measured by dynamic light scattering instrument (Zetasizer ELSZ-200ZS, Malvern Instruments Ltd.).

## Immunogenicity assessment in mouse models

The immunogenicity studies in mice were conducted at Noble Life Sciences Inc, (Sykesville, MD) or the National Institutes of Biomedical Innovation, Health and Nutrition (NIBIOHN; Osaka, Japan). Female 6- to 8-week-old BALB/c mice ($n = 5$ per group) were housed with room temperature maintained between 65-72 °F with 30-70% relative humidity and 12:12 light/dark cycle. Animals were injected intramuscularly with the immunogen in 50 μL of formulation buffer. An equivalent volume of phosphate buffered saline (PBS) was injected for the control group. For initial down-selection of the RBD variants, VEEV-VRP encapsulated RBD variants ($2.2 \times 10^6$ – $1 \times 10^7$ IU) were injected at weeks 0 and 4, or three times at weeks 0, 1 and 2. The control group received 10 μg of SARS-CoV-2 (2019-nCoV) Spike S1-His recombinant protein (Sino Biological) with aluminum hydroxide adjuvant (Alhydrogel 2%, InvivoGen). For subsequent immunogenicity and cross reactivity determinations, saRNA RBD variants were formulated with LNP, and animals were immunized with 1-10 μg of formulated RNA at weeks 0 and 4. Mouse sera were collected at weeks 0, 2, 4, 6 and 8. Blood cell counts were monitored using an automated hematology analyzer VETSCAN HM5 (Zoetis).

## Immunization/challenge study in hamster model

The hamster studies were conducted at Bioqual Inc. (Rockville, MD, USA). Golden Syrian hamsters ($n = 6$ per group, three male and three female, 6–8 weeks old) were housed at Bioqual with room temperature maintained between 68-79 °F with 30-70% relative humidity and 12:12 light/dark cycle. Animals were anesthetized, and then immunized twice with 10 μg of saRNA-RBD (Wuhan-Hu-1 or Gamma)-TM in 50uL of formulation buffer at weeks 0 and 4. Hamster sera were collected at weeks 0, 2, 4, 6 and 8 (day 55). At day 56, the immunized hamsters were challenged intranasally with SARS-CoV-2 Wuhan-Hu-1 (USA-WA1/2020, lot# 12152020-1235) or Gamma (lot# 031921-1215) in a biosafety level 3 facility. Body weight changes for each hamster were measured daily post challenge. RT−qPCR was performed on days 2, 4, 7 and 10 post challenge. Tissue samples were collected on day 10 post challenge.

## Immunization and Challenge studies in non-human primate model

Eighteen cynomolgus macaques were singly housed at NIBIOHN. The ambient temperature was maintained between 20–25 °C with 12:12 light/dark cycle. Cynomolgus macaques ($n = 6$ per group, 2 male and 4 female, 6 to 20 years old) were immunized twice with saRNA-LNP expressing SARS-CoV-2 RBD(Wuhan-Hu-1)-TM or (Gamma)-TM intramuscularly at a 4-week interval. Blood was collected every other week from −2 to 28 weeks post immunization. The lymph nodes were collected at −2 and 6 weeks post immunization. PBMCs and plasma were isolated from blood via density gradient centrifugation using Ficoll-Paque Plus (Cytiva). PBMCs were immersed in fetal bovine serum containing 10% DMSO and stored at −150 °C until analysis. The NHP challenge study was performed at Bioqual Inc. Adult cynomolgus macaques ($n = 5$ per group, two male and three female, 2–4 years old) were pair-housed within study groups at Bioqual at the temperature between 64–84 °F with 30–70% relative humidity and 12:12 light/dark cycle. Animals were injected intramuscularly twice with 50 μg of saRNA RBD(Wuhan-Hu-1)-TM at 0 and 4 weeks. The blood samples were collected at 56 days post immunization and the serum binding antibody titers and ACE2 binding inhibition activity against SARS CoV-2 RBD variants (Wuhan-Hu-1, Beta, Gamma, and Delta) were assessed. The immunized monkeys were challenged with SARS-CoV-2 via the combination intranasal/intratracheal (IN/IT) route at Day 56 post immunization. BAL and nasal/oropharyngeal swabs were collected from challenged animals at 2 and 4 days post challenge, and the level of infectious virus was assessed by $TCID_{50}$.

## Infectious Viral Load Assay ($TCID_{50}$)

Vero TMPRSS2 cells (obtained from the Vaccine Research Center-NIAID) were plated at 25,000 cells/well in DMEM + 10% FBS + Gentamicin and the cultures were incubated at 37 °C, 5.0% CO2. Cells were 80–100% confluent the following day. Medium was aspirated and replaced with 180 μL of DMEM + 2% FBS + gentamicin. Twenty (20) μL of sample was added to the top row in quadruplicate and mixed using a P200 pipettor five times. Twenty μL were then transferred to the next row, and repeated down the plate (columns A-H) representing 10-fold dilutions. Positive (virus stock of known infectious titer in the assay) and negative (medium only) control wells were included in each assay. The plates were incubated at 37 °C, 5.0% CO2 for 4 days. The cell monolayers are visually inspected for CPE. Non-infected wells will have a clear confluent cell layer while the infected cells show cell rounding. The $TCID_{50}$ value is calculated using the Read-Muench formula. For samples which have <3 CPE positive wells, the $TCID_{50}$ cannot be calculated using the Reed-Muench formula. These samples are assigned a titer of below the limit of detection (i.e., <2.7 log10 $TCID_{50}$/mL).

## Enzyme-Linked Immunosorbent Assay (ELISA)

ELISA assays were used to evaluate anti-SARS-CoV-2 antibodies in immunized mice, hamsters and monkeys. Nunc-Immuno MicroWell 96-well plates (Thermo Fisher Scientific) were coated with recombinant SARS-CoV-2 (2019-nCoV) Spike S1, RBD or variant RBD proteins (Sino Biological or Acro Biosystems) at 0.1 μg/mL or 0.5 μg/mL in PBS overnight at 4 °C. Coated plates were washed three times with Tris-buffered saline with 0.05% Tween-20 (TBS-T) before blocking with blocking buffer (5% non-fat milk in TBS-T) for 1 h at ambient temperature. After blocking, serum or plasma samples were plated at a 1:50 initial dilution, followed by 3 or 4-fold serial dilution in blocking buffer. The plates were washed again three times before addition of 100 μL of the appropriate HRP-conjugated secondary antibody (1:4000 dilution, Santa Cruz). After 1 h-incubation at room temperature, the plates were washed, developed using 3, 3', 5, 5'-tetramethylbenzidine (TMB) Microwell Peroxidase Substrate (Seracare) for 20 min, and stopped by 2 N sulfuric acid in water (Avantor). The absorbance values at 450 nm ($OD_{450}$) were read in a BioTek Synergy HTX Multi-Mode reader (Agilent). Endpoint tiers were defined as the $log_{10}$ dilution resulted in an $OD_{450} = 1.0$ or 0.3 using 4PL regression and Prism 9.5.1. software (GraphPad). Serum titers with undetectable binding were assigned the value one-half lower limit of detection (LLOD), while those over $OD_{450} = 1.0$ were assigned the value upper limit of detection (ULOD). The First WHO international Standard for anti-SARS-CoV-2 immunoglobulin was obtained from NIBSC (NIBSC code: 20/136). The Standard has been evaluated in a WHO International Collaborative study[30].

## ELISA-based competitive surrogate neutralization assay

The ability of induced antibodies to inhibit the binding of RBD to Angiotensin-converting enzyme 2 (ACE2) was evaluated using the SARS-CoV-2 Surrogate Virus Neutralization Test kit (GenScript). Serum samples were diluted at a 1:10 initial dilution, followed by 5-fold serial dilution, in sample dilution buffer, and then mixed at a 1:1 volume ratio with HRP-conjugated RBD solution that was diluted at 1:999 with RBD dilution buffer according to the manufacturer's protocol. The recombinant variant RBD proteins labeled with HRP (GenScript) were

purchased separately and tested at the same procedure as the Wuhan-Hu-1 RBD protein. A mixture of the serum sample and the HRP-conjugated RBD protein was incubated at 37 °C for 30 min, and 100 μL of the mixture was transferred to the ACE2 coated well. Plates were incubated at 37 °C for 15 min, washed 4 times with 260 μL of wash buffer, and developed using TMB in the dark for 20 min. After addition of the stop solution, the $OD_{450}$ values were measured in the BioTek Synergy HTX Multi-Mode reader. We confirmed that the $OD_{450}$ values of the positive and negative controls were in the range of <0.3 and > 1.0, respectively. The percent of ACE2 inhibition was calculated as $(1 − OD_{450}$ of the sample)/($OD_{450}$ value of negative control) × 100. For statistical analysis, calculated $log_{10}$ ($ID_{50}$) values higher than ULOD were assigned a value of the ULOD, and calculated values lower than the LLOD were assigned the value of one-half of the LLOD.

### Neutralization assay of live SARS-CoV-2

Neutralization assays were performed as previously described[31]. Briefly, VeroE6/TMPRSS2 cells were maintained in Dulbecco's modified Eagle medium (DMEM) (Thermo Fisher Scientific or Sigma-Aldrich) supplemented with 10% fetal bovine serum (FBS) and 1% Penicillin and Streptomycin (Thermo Fisher Scientific) (hereinafter referred to as GM) at 37 °C in a 5% $CO_2$ incubator, and were seeded into 96-well plates at $3 ×10^3$ cells/well (100 μL/well) on the day prior to infection. Sera were serially diluted in GM, mixed with each SARS-CoV-2 strain at 2,000 $TCID_{50}$/mL (final concentration), and incubated at 37 °C for 60 min. The serum-virus mixture (100 μL/well) was directly applied to VeroE6/TMPRSS2 cells for infection, and incubated at 37 °C for 60 min. The cells were washed once with GM, and further incubated in fresh GM (100 μL/well) at 37 °C. At 20 h post infection, culture supernatants were harvested for quantifying viral amounts. 2.5 μL of the supernatants was used to quantify the SARS-CoV-2 RNA copies by RT-qPCR method using the PrimeDirect Probe RT-qPCR Mix (Takara Bio) and the primer/probe set for N2. The RT-qPCR was performed in the Thermal Cycler Dice Real Time System III (Takara Bio) according to the default condition of the manufacture's protocol. RNA copy numbers were calculated relative to a four-point standard curve from the virus stock whose RNA concentration has already been determined by the RT-ddPCR. All statistical analyses were performed using Prism 9.5.1 software (GraphPad).

### Neutralization assay of Pseudotyped virus

For determination of the pseudovirus neutralization titer (pVNT) of vaccinated monkey plasma, HEK-293A expressing ACE2 and TMPRSS2 were seeded in 96-well plates. After 24 hr, the monkey plasma was serially diluted in 3-fold increments, starting at 1:3, and incubated with SARS-CoV-2 pseudotyped virus (expressing luciferase) at 37 °C for an hour. After incubation, mixture of plasma and pseudotyped virus was added to each well. After 24 hr, the luciferase activity was measured using Enspire (PerkinElmer). The pVNT50 was defined as plasma dilution achieved 50% inhibition of pseudotyped virus infection using Prism 9.5.1. software (GraphPad).

### Calculation of antibody half-life time

Antibody half-lives at a slow decay period in macaque model were calculated using power-law model, by fitting titers starting at the week 12 after the first dose vaccination and beyond. The lme4 package in R.4.0.4 (https://www.R-project.org/) was applied for the calculation as previously reported[32,33]. Briefly, the formula of the power-law model was taken from the following formula:

$$log_{10}(Titer_{i,j}) = (\alpha + a_i) + \beta \times [log_{10}(study\,week_{i,j} - 4)] + e_{i,j}$$

where $\alpha$ and $\beta$ are fixed-effect intercept and decay rate. $a_i$ and $e_{i,j}$ are the random intercept and the model error for macaque $i$ at study week $j$, respectively. To account for the second dose vaccination at the week

4, our study week was offset by 4. The half-life ($t_{1/2}$) was calculated based on the following formula:

$$log_{10}(t_{1/2}) = log_{10}(study\,week − 4) + [log_{10}(0.5)]/\beta$$

In this study, the half-life at the week 28 for the power-law model was determined, because the week 28 was an approximate midpoint between the starting (week 12) and the last time points (week 52) of the study.

### Statistical analysis

ELISA titers ($OD_{450}$) and ACE2 inhibition (%) are shown as means ± standard error of the mean (sem). Individual endpoint titers or $ID_{50}$ values are shown with the calculated median. Statistical analyses were performed using Prism 9.5.1 software (GraphPad). Mann–Whitney test, Wilcoxon matched-pairs signed-rank test were used for comparisons of two groups. One-way analysis of variant (ANOVA) was used for comparisons across multiple groups. Correlations were calculated using a nonparametric Spearman's rank test. $P$-values of <0.05 were considered statistically significant.

### Ethics statement

The mice and hamster studies were conducted without direct involvement of Quality Assurance but adhered to applicable Noble Life Sciences, Inc (Maryland, US) or Bioqual (Maryland, US) SOPs. This study was conducted in compliance with the current version of the following (1) Animal Welfare Act Regulations (9 CFR); (2) U.S. Public Health Service Office of Laboratory Animal Welfare (OLAW) Policy on Humane Care and Use of Laboratory Animals; (3) Guide for the Care and Use of Laboratory Animals (Institute of Laboratory Animal Resources, Commission on Life Sciences, National Research Council, 1996); and (4) AAALAC accreditation. These studies were approved by Noble Life Sciences Inc IACUC committee (Approved no. NLS-591 and NLS-631) and Bioqual IACUC committee (Approved no. 20-010 and 20-145). NHP studies were performed at NIBIOHN with approval from the Committee on the Ethics of Animal Experiments of NIBIOHN (Approved no. #DSR03-25).

### Reporting summary

Further information on research design is available in the Nature Portfolio Reporting Summary linked to this article.

## Data availability

Datasets generated and/or analysed during the current study are available in the paper or are appended as supplementary data. The source data underlying Figs. 1–5 have been provided as a Source Data file. Source data are provided with this paper.

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

## Acknowledgements

This research was supported by AMED under Grant Number JP21nf0101627 (T.N., T.S., Y.M., H.O., T.T., R.S., Y.K., T.F., Y.I., T.Y. and W.A.). We thank member of Japan Agency for Medical Research and Development (AMED) in particularly, K. Sudo, A. Fujie, K. Arai, T. Fukaya, R. Itaya and N. Chiba for helpful discussion and supports. We also thank N. Uemura, S. Tsujihata, W. Sugiura, N. Sato, M. Nakata and M. Tada for advice and thank K. Swagata, H. Andersen, S. Tati for arranging animal experiments.

## Author contributions

Conceived study: T.S., J.A., T.F., Y.I., T.Y., J.F.S. and W.A. Performed experiments: M.K., T.N., A.L.M., K.I., M.R.H. Y.M., T.T., R.S. and Y.K. Performed statistical analysis: M.K., T.N., H.O., Y.I. and T.Y. Wrote manuscripts: M.K., T.N., K.M., Y.I., T.Y., J.F.S. and W.A.

## Competing interests

M.K., A.L.M., K.I. and K.M. are employees (VLP Therapeutics, Inc.); W.A. is a board member, an employee and holds stocks (VLP Therapeutics, Inc.); J.F.S. is an employee and holds stocks (VLP Therapeutics. Inc.); J.A. and M.R.H. were employees at VLP Therapeutics. T.S. is an employee (VLP Therapeutics Japan, LLC); W.A. is a management board member (VLP Therapeutics Japan, LLC); W.A. and J.F.S. are inventors on related vaccine patents (Coronavirus vaccine #17/232666 and Efficient vaccine #18/067358). The remaining authors declare no competing interest.
