## [Peer Review File · Nature Communications]

saRNA Vaccine Expressing Membrane-Anchored RBD Elicits Broad and Durable Immunity Against SARS-CoV-2 Variants of ConcernReviewers' Comments:

Reviewer #1:

Remarks to the Author:

Overview:

The study by Komori et al reports the immunogenicity of an alphavirus (VEEV)-based self-amplifying RNA vaccine expressing an RBD membrane-anchored by fusing an N-terminal sequence and a C-terminal transmembrane domain from influenza hemagglutinin protein (referred to as RBD-TM) to display the RBD sequence on the surface of transfected cells. The vaccine was delivered IM in lipid nanoparticles. An interesting finding from this paper is that the RBD-TM saRNA vaccine induced higher IgG antibody and neutralizing antibody against the original (alpha?) variant as well as cross-reactive responses against Gamma, Delta and Omicron (BA.1) variants when compared to the secreted RBD and a recombinant S1 protein formulated in ALUM. The RBD-TM saRNA vaccine also induced higher Th1 responses when compared to the other modalities as indicated by a higher IgG2a/IgG1 ratio and higher frequencies of B cell memory and notably, data mice shows that mice immunized with the RBD-TM vaccine had lower WBC counts and granulocytes than mice immunized with the secreted form, suggesting lower reactogenicity. In support of these data, the authors also show that the stronger Th1 responses induced by the saRNA RBD-TM vaccine correlated with stronger antibody responses. Another interesting finding is that an RBD-TM saRNA vaccine based on Gamma sequences induced broader antibody responses than one based on the original D614G strain. They also show protection from SARS-CoV-2 infection in nonhuman primates challenge 2 months after the final dose. The authors also report induction of sustained binding antibody and central memory T cell responses in nonhuman primates for up to 6 months. While the data showing improved immunogenicity with the membrane-anchored RBD when compared to the secreted RBD is intriguing, a major weakness of this study is that it lacks the appropriate controls to discern if the improved immunogenicity relative to the control mRNA vaccine expressing full-length Spike (S2P) is due primarily to the immunogen or the self-amplifying RNA platform. In addition, the data supporting the premise that this vaccine could offer more prolonged immunity compared to mRNA vaccines is very thin as the authors did not evaluate durability of neutralizing antibody responses or protective efficacy past 2 months. A detailed summary of these and other major and minor weaknesses is as follows:

MAJOR

1. No direct comparison of RBD-TM vs S2P using same platform technology: The authors nicely establish that the RBD-TM saRNA vaccine is more immunogenic in the induction of antibody and polyfunctional T cell responses than the secreted RBD and a full length Spike (S2P) protein expressed by mRNA vaccine (which models current COVID19 mRNA vaccines). However, from these data it is not clear if the higher immunogenicity observed with the RBD-TM saRNA vaccine as compared to the mRNA S2P vaccine is due primarily to the self-amplifying RNA platform or the RBD-TM immunogen. A direct comparison of the RBD-TM to the S2P using same vaccine platform (i.e using saRNA) is needed to determine if the RBD-TM immunogen may be a superior immunogen when compared to pre-fusion Spikes

2. The conclusion that this vaccine induces durable immunity is weak: While the authors show sustained binding antibody in nonhuman primates for 6 months post-immunization, this is not sufficient data to support the conclusion that they have achieved "prolonged immunity". Notably, a previously study showed that Moderna's mRNA vaccine induces sustained binding antibody for 1 year but this did not predict durability of neutralizing antibody that waned more quickly (<https://www.ncbi.nlm.nih.gov/pmc/articles/PMC8639396/pdf/main.pdf>). To support durability, the authors should also evaluate neutralizing antibody over the 6 month time period. Furthermore, it should be noted that in this previous paper, the Moderna mRNA vaccines afforded durable protective efficacy in nonhuman primates 12 months post-immunization and this level of protection was comparable to what the authors report the RBD-TM saRNA vaccine afforded in macaques challenged only 2 months post-immunization. Based on this comparison, the implication the authors make in the conclusion and in the manuscript title that this vaccine may have an advantage in the induction of

more durable immunity than currently licensed mRNA vaccines is not sufficiently supported by the data or prior published results. The authors should more precisely address how their results compare to previous published results with the Moderna and Pfizer vaccines showing durability of immune responses and protection and acknowledge the limitations of their study, namely that additional studies, including analysis of neutralizing antibody and a challenge at 6+ months will be needed to determine if the RBD-TM saRNA vaccine provides an advantage in durability of immunity compared to currently licensed mRNA vaccines. Without stronger data supporting durability, the title of this paper is misleading and should be changed.

3. No analysis of neutralizing antibody responses vs Omicron variants in NHPs. This is needed to bring this paper up to current state of art.

4. It's interesting that the RBD-TM Gamma vaccine afforded better cross-protection vs both WT and Gamma whereas the WT vaccine was less effective vs Gamma. Can the authors speculate on the reason for this outcome? Given that the Gamma vaccine also induced stronger cross-reactive antibody vs Delta and Omicron, challenges vs these more contemporary strains (Delta and Omicron) would increase the impact of this observation.

MINOR

1. The authors refer to the original D614G variant as "Wild Type" or WT. This is confusing as WT usually refers to a virus found in nature and all variants are found in nature. The original stain (i.e. prior to Alpha strain or B.1.1.7 strain) is more accurately referred to as the WA-1, Wuhan or D614G strain.

2. The authors should define, within the results section (not just the methods) what is the "WHO standard sera" is.

3. Lines 272-273: This sentence does not seem to be complete.

4. Lines 305-307: The authors cite a study of the Pfizer mRNA vaccine exhibiting a half-life of 42 days for antibody between months 3-6 and then claim that in contrast, they did not see a significant decay in RBD-specific antibodies with the saRNA RBD-TM vaccine. They use this information to support the idea that their vaccine may offer more durable immunity. To better support this claim, the authors should similarly calculate the decay rate in RBD-specific antibody in their study between months 3-6 and offer that number as a comparison. They should also acknowledge the Pfizer mRNA vaccine report also showed slower decay of the neutralizing antibody.

Reviewer #2:

Remarks to the Author:

This is an interesting study in which a single-cycle vector system utilizing a Venezuelan Equine Encephalitis Virus-based replicon expression vector expressing the alphavirus nsPs1-4 replicating and transcribing the saRNA drives production of secreted RBD and membrane-anchored RBD (RBD-TM). While it does indeed prove interesting, the rationale for adding in the 3 Gamma mutations is a little opaque – was the logic that this might be a step to a pan-VOC spike protective sequence?

There is a nicely comprehensive work package to evaluate the saRNA vaccines through Balb/c, hamster and macaque models, with interesting findings.

In Fig 2F it was unclear precisely what are the units for the axis?

Live virus neutralization is mentioned in the Methods – I may have missed the figure in the Supplementary data, but surely, apart from the challenge data, the thing that one would most like to see is the comparative timecourse for neutralization IC50s across each of the variants.

I appreciate the manuscript may have been written prior to the dominance of BA.5 (and then, BQ1.1),

but it would be valuable to see cross-protection evaluated against these most concerning current VOC.

Overall, an interesting paper offering some useful increment. My only real niggle is that the paper seeks to make big claims about broader, more durable responses, yet the claim is hard to truly evaluate as we never see a long timeline (beyond 28 weeks) with a side by side comparison to one of the current mRNA vaccines.

Reviewer #1 (Remarks to the Author):

Overview:

The study by Komori et al reports the immunogenicity of an alphavirus (VEEV)-based self-amplifying RNA vaccine expressing an RBD membrane-anchored by fusing an N-terminal sequence and a C-terminal transmembrane domain from influenza hemagglutinin protein (referred to as RBD-TM) to display the RBD sequence on the surface of transfected cells. The vaccine was delivered IM in lipid nanoparticles. An interesting finding from this paper is that the RBD-TM saRNA vaccine induced higher IgG antibody and neutralizing antibody against the original (alpha?) variant as well as cross-reactive responses against Gamma, Delta and Omicron (BA.1) variants when compared to the secreted RBD and a recombinant S1 protein formulated in ALUM. The RBD-TM saRNA vaccine also induced higher Th1 responses when compared to the other modalities as indicated by a higher IgG2a/IgG1 ratio and higher frequencies of B cell memory and notably, data mice shows that mice immunized with the RBD-TM vaccine had lower WBC counts and granulocytes than mice immunized with the secreted form, suggesting lower reactogenicity. In support of these data, the authors also show that the stronger Th1 responses induced by the saRNA RBD-TM vaccine correlated with stronger antibody responses. Another interesting finding is that an RBD-TM saRNA vaccine based on Gamma sequences induced broader antibody responses than one based on the original D614G strain. They also show protection from SARS-CoV-2 infection in nonhuman primates challenge 2 months after the final dose. The authors also report induction of sustained binding antibody and central memory T cell responses in nonhuman primates for up to 6 months. While the data showing improved immunogenicity with the membrane-anchored RBD when compared to the secreted RBD is intriguing, a major weakness of this study is that it lacks the appropriate controls to discern if the improved immunogenicity relative to the control mRNA vaccine expressing full-length Spike (S2P) is due primarily to the immunogen or the self-amplifying RNA platform. In addition, the data supporting the premise that this vaccine could offer more prolonged immunity compared to mRNA vaccines is very thin as the authors did not evaluate durability of neutralizing antibody responses or protective efficacy past 2 months. A detailed summary of these and other major and minor weaknesses is as follows:

MAJOR

Reviewer 1:

1. No direct comparison of RBD-TM vs S2P using same platform technology: The authors nicely establish that the RBD-TM saRNA vaccine is more immunogenic in the induction of antibody and polyfunctional T cell responses than the secreted RBD and a full length Spike (S2P) protein expressed by mRNA vaccine (which models current COVID19 mRNA vaccines). However, from these data it is not clear if the higher immunogenicity observed with the RBD-TM saRNA vaccine as compared to the mRNA S2P vaccine is due primarily to the self-amplifying RNA platform or the RBD-TM immunogen. A direct comparison of the RBD-TM to the S2P using same vaccine platform (i.e using saRNA) is needed to determine if the RBD-TM immunogen may be a superior immunogen when compared to pre-fusion Spikes.

Response:

As suggested by the reviewer, we have performed a mouse *in vivo* immunogenicity study to directly compare the RBD-TM and S2P antigen on the same saRNA platform. The immunogenicity of RBD-TM was significantly greater than that of S2P in both 1µg and 10µg/dose groups. We have added Extended Data Figure 2f and included the following statement in the result section (Lines 146-151 in the track change version)

“We also directly compared the immunogenicity of RBD-TM and whole S(Wuhan) as a prefusion form (S2P) using the same saRNA platform. We immunized mice with either 1 or 10µg of saRNA expressing RBD-TM or S2P. The magnitude of binding antibodies was greater in RBD-TM immunized mice when compared to pre-fusion S2P immunogen (**Extended Data Fig. 2f**), suggesting that the RBD based design can be an alternative candidate of SARS-CoV-2 vaccine immunogen.”

Reviewer 1:

2. *The conclusion that this vaccine induces durable immunity is weak: While the authors show sustained binding antibody in nonhuman primates for 6 months post-immunization, this is not sufficient data to support the conclusion that they have achieved “prolonged immunity”. Notably, a previously study showed that Moderna's mRNA vaccine induces sustained binding antibody for 1 year but this did not predict durability of neutralizing antibody that waned more quickly (<https://www.ncbi.nlm.nih.gov/pmc/articles/PMC8639396/pdf/main.pdf>). To support durability, the authors should also evaluate neutralizing antibody over the 6 month time period. Furthermore, it should be noted that in this previous paper, the Moderna mRNA vaccines afforded durable protective efficacy in nonhuman primates 12 months post-immunization and this level of protection was comparable to what the authors report the RBD-TM saRNA vaccine afforded in macaques challenged only 2 months post-immunization. Based on this comparison, the implication the authors make in the conclusion and in the manuscript title that this vaccine may have an advantage in the induction of more durable immunity than currently licensed mRNA vaccines is not sufficiently supported by the data or prior published results. The authors should more precisely address how their results compare to previous published results with the Moderna and Pfizer vaccines showing durability of immune responses and protection and acknowledge the limitations of their study, namely that additional studies, including analysis of neutralizing antibody and a challenge at 6+ months will be needed to determine if the RBD-TM saRNA vaccine provides an advantage in durability of immunity compared to currently licensed mRNA vaccines. Without stronger data supporting durability, the title of this paper is misleading and should be changed.*

Response:

We appreciate the reviewer’s comments. We performed binding/neutralizing antibody assay at 52 weeks and confirmed that these antibodies induced by saRNA RBD(Gamma)-TM was maintained even for 52 weeks post immunization. We did not generate a challenge at 6+ month data due to difficulties in obtaining more NHP for this study.

We have included the later time point (52wk) to **Fig. 6b** and generated a new figure (**Fig. 6d**) showing the neutralizing antibody titers in saRNA RBD(Gamma)-TM immunized NHPs. We

have introduced the following statement in result section. (Lines 270-276 in the track change version)

“Neutralization activity was also assessed to confirm that it is maintained throughout the observation period (52 weeks). The neutralization activity of the serum was tested against the pseudotyped virus expressing the S protein autologous to immunogen (**Fig. 5d**). As expected, we detected significant differences in neutralizing antibody titer between saRNA RBD (Wuhan)-TM and saRNA RBD (Gamma)-TM at weeks 28 ($p=0.05$) and 52 ($p=0.05$). The neutralizing antibody titer of saRNA RBD (Gamma)-TM animals at 52 weeks was comparable level with that of 28 weeks time point, indicating durable neutralizing antibody response was induced.”

In addition, as the reviewer suggested, we modified the title. We have replaced the word “prolonged” to “durable” and excluded the word “protection”. Our data suggest durability in saRNA RBD(Gamma)-TM induced binding and neutralizing antibodies up to 52 weeks post immunization. These modifications would make the title more consistent with the data sets we are presenting in this current study.

Reviewer1:

3. No analysis of neutralizing antibody responses vs Omicron variants in NHPs. This is needed to bring this paper up to current state of art.

Response:

We agree to the reviewer’s point, and we assessed the serum neutralizing activity against the live Omicron variant (BA.5). Unfortunately, we couldn’t detect neutralization activity against BA.5 at 52 wks post immunization.

Reviewer1:

4. It’s interesting that the RBD-TM Gamma vaccine afforded better cross-protection vs both WT and Gamma whereas the WT vaccine was less effective vs Gamma. Can the authors speculate on the reason for this outcome? Given that the Gamma vaccine also induced stronger cross-reactive antibody vs Delta and Omicron, challenges vs these more contemporary strains (Delta and Omicron) would increase the impact of this observation.

Response:

We thank the reviewer for raising the question on the inherent characteristic of Gamma antigen which induces better antibody response compared to the Wuhan antigen. Although we do not have the direct data nor able to conduct the challenge study against more contemporary variants, one can speculate that mutations such as K417T may be conferring the stability of the RBD (Starr *et al.*; doi.org/10.1016/j.cell.2020.08.012), which may contribute to induction of better antibody responses. We will explore the mechanisms in future studies.

MINOR

Reviewer 1:

5. The authors refer to the original D614G variant as “Wild Type” or WT. This is confusing as WT usually refers to a virus found in nature and all variants are found in nature. The original strain (i.e. prior to Alpha strain or B.1.1.7 strain) is more accurately referred to as the WA-1, Wuhan or D614G strain.

Response:

We thank the reviewer for noting this point. As the reviewer suggested, we corrected and referred to the original strain as Wuhan and original D614G variant as D614G in text and Figures.

Reviewer 1:

6. The authors should define, within the results section (not just the methods) what is the “WHO standard sera” is.

Response:

We thank the reviewer for noting this point and we defined the WHO standard sera in the results section as follows (Lines 251 – 253 in the track change version).

“The standard WHO sera was included in the panel both as an internal control and as a standardized control to allow direct comparison of our results with the datasets from other groups.”

We also included the following statement in the figure legend of Figure 5:

“The standard WHO sera is 1000 binding antibody units (BAU)/mL”

Reviewer 1:

7. Lines 272-273: This sentence does not seem to be complete.

Response:

We thank the reviewer for spotting the error in the manuscript. We have made the following modification to complete the sentence (Lines 289-292 in the track change version).

“Here, antigen-specific CD4 T cells were defined as CD4⁺ total memory T cells expressing CD154, and significant frequencies of antigen-specific CD4 T cells were induced (**Fig. 5f left panel**), and there was significant induction of Th1 phenotype CD4 memory T cells (**Fig. 5f middle and right panel**).”

Reviewer 1:

8. Lines 305-307: The authors cite a study of the Pfizer mRNA vaccine exhibiting a half-life of 42 days for antibody between months 3-6 and then claim that in contrast, they did not see a significant decay in RBD-specific antibodies with the saRNA RBD-TM vaccine. They use this information to support the idea that their vaccine may offer more durable immunity. To better support this claim, the authors should similarly calculate the decay rate in RBD-specific antibody in their study between months 3-6 and offer that number as a comparison. They should also acknowledge the Pfizer mRNA vaccine report also showed slower decay of the neutralizing antibody.

Response:

We appreciate reviewers for raising the point regarding the half-life of antibody responses induced by immunization of saRNA RBD vaccine. We have calculated the half-life using power-law models (this model assumes that decay rates decrease over time) by fitting binding antibody titers between months 3-7, and included the following statement in the result section (Lines 257-263 in the track change version):

“Next, we have calculated the estimated half-life of binding antibody response in NHPs immunized with saRNA RBD(Wuhan)-TM and saRNA RBD(Gamma)-TM between months 3-7 against Wuhan and Gamma antigens. The estimated half-life of binding antibody response induced by saRNA RBD(Wuhan)-TM was 178.5 days against Wuhan and 149.8 days against Gamma, whereas the half-life of binding antibody response induced by saRNA RBD(Gamma)-TM was 198.8 days against Wuhan and 112.7 days against Gamma, calculated with the use of power-law model.”

Since the experimental settings are variable between studies (e.g., human vs NHP, time points used, etc.), it would be difficult to make a direct comparison of half-life between our candidate vaccine and others (mRNA). Thus, we have reoriented our point of discussion to focus primarily on the saRNA platform and the antigen design. We have introduced changes in the discussion section. Together with the modifications of the study title from reviewer’s earlier comment, we believe that this will avoid delivering the misleading message to the readers that saRNA vaccines are superior to mRNA vaccines.

Reviewer #2 (Remarks to the Author):

This is an interesting study in which a single-cycle vector system utilizing a Venezuelan Equine Encephalitis Virus-based replicon expression vector expressing the alphavirus nsPs1-4 replicating and transcribing the saRNA drives production of secreted RBD and membrane-anchored RBD (RBD-TM).

Reviewer 2:

1. While it does indeed prove interesting, the rationale for adding in the 3 Gamma mutations is a

little opaque – was the logic that this might be a step to a pan-VOC spike protective sequence? There is a nicely comprehensive work package to evaluate the saRNA vaccines through Balb/c, hamster and macaque models, with interesting findings.

Response:

We thank the reviewer for the comment on our manuscript. Although our data was convincing that RBD sequence derived from the Gamma variant induces broader and more potent antibody responses against VOCs compared to the RBD sequence derived from Wuhan, future studies including clinical trials may give us a direct answer. We are currently preparing a clinical trial testing saRNA expressing this Gamma RBD-TM antigen.

Reviewer 2:

2. In Fig 2F it was unclear precisely what are the units for the axis?

Response:

The Y-axis represents the frequency of each subset listed on the X-axis among CD4 total memory T cells that produce IFN γ , TNF, or IL-2. We have corrected the axis label from "% of cytokine-producing CD4 T cells" to "% of each subset in cytokine-producing CD4 T cells". (See new Figure 2f). Furthermore, we have similarly corrected the axis-label of Figure 2i (See new Figure 2i).

Reviewer 2:

3. Live virus neutralization is mentioned in the Methods – I may have missed the figure in the Supplementary data, but surely, apart from the challenge data, the thing that one would most like to see is the comparative timecourse for neutralization IC50s across each of the variants.

Response:

We appreciate the reviewer's suggestion. We have generated a new figure (Fig. 6d) showing the maintenance and time course of neutralizing antibody titers in saRNA RBD(Gamma)-TM immunized animals. We have introduced the following statement in result section (Lines 270-276 in the track change version).

“Neutralization activity was also assessed to confirm that it is maintained throughout the observation period (52 weeks). The neutralization activity of the serum was tested against the pseudotyped virus expressing the S protein autologous to immunogen (Fig. 5d). As expected, we detected significant differences in neutralizing antibody titer between saRNA RBD (Wuhan)-TM and saRNA RBD (Gamma)-TM at weeks 28 ($p=0.05$) and 52 ($p=0.05$). The neutralizing antibody titer of saRNA RBD (Gamma)-TM animals at 52 weeks was comparable level with that of 28 weeks time point, indicating durable neutralizing antibody response was induced”

Reviewer 2:

4. I appreciate the manuscript may have been written prior to the dominance of BA.5 (and then, BQ1.1), but it would be valuable to see cross-protection evaluated against these most concerning current VOC.

Response:

We agree to the reviewer's point, and we assessed the serum neutralizing activity against the live BA.5 variant. Unfortunately, we could not detect neutralization activity against BA.5 at 52 wks post immunization (data not shown). Due to the lack of neutralization activity against BA.5, we did not test against BQ1.1.

Reviewer2 :

5. Overall, an interesting paper offering some useful increment. My only real niggle is that the paper seeks to make big claims about broader, more durable responses, yet the claim is hard to truly evaluate as we never see a long timeline (beyond 28 weeks) with a side by side comparison to one of the current mRNA vaccines.

Response:

We thank the reviewer for noting this point and have revised the text accordingly. We included binding antibodies data at additional time point (52 wks) beyond 28 wks (Figure 5b) and the new neutralizing antibodies data (Figure 5d) showing that saRNA RBD(Gammar)-TM induced durable antibody responses even at 52 wks. These additional data, together with the calculation of the half-life of binding antibody titers (see our response to reviewer1) suggest that saRNA induced durable immune responses. However, because we did not make head-to-head comparison to the current mRNA vaccines in this study, we have modified the discussion section to exclude the statements of direct comparison of our saRNA vaccine and mRNA vaccines.

We have recently conducted a randomized, observer-blind, phase 1 study of saRNA RBD (Wuhan)-TM vaccine (https://papers.ssrn.com/sol3/papers.cfm?abstract_id=4309025). In this study, we demonstrated side by side comparison between saRNA RBD(Wuhan)-TM and mRNA vaccine (BNT162b2). Our data suggest that saRNA RBD (Wuhan)-TM was able to induce durable antibody responses even up to 28 wks. Phase 2 study for saRNA RBD(Wuhan)-TM is now in progress. Furthermore, phase 1/2 study for saRNA RBD(Gammar)-TM with a side by side comparison to current mRNA vaccines will start in April 2023. Based on this pre-clinical study, we hope that the breadth and the durability of the antibody responses induced by saRNA RBD(Gammar)-TM might be observed in the clinical trials.

\Reviewers' Comments:

Reviewer #1:

Remarks to the Author:

The reviewers revised manuscript addresses most of the concerns. In particular, the decision to remove statements of comparison between their saRNA and mRNA vaccines given that the study did not directly compare the two is appropriate.

A lingering issue is the concern raised by both reviewers about testing neutralization vs more contemporary Omicron strains (i.e. BA5). In the response to the reviewers, the authors state they analyzed neutralizing activity vs BA5 but did not detect significant activity. However, it appears this information has not been incorporated anywhere in the manuscript. "Data not shown" is not really acceptable in this case. The authors should specifically state in the paper that they did not see neutralizing responses vs BA5 and show that data in Fig 4 alongside the positive neutralizing data they showed vs other VOCs or in a supplemental figure.

Reviewer #2:

Remarks to the Author:

The authors have responded fairly comprehensively to a strong set of referee reports and produced an improved manuscript.

The lack of detectable neutralisation of Omicron sub-variants is quite a strong caveat to their observations and claims - it would be sensible for them to allocate greater prominence to this point

Reviewer 1 :

The reviewers revised manuscript addresses most of the concerns. In particular, the decision to remove statements of comparison between their saRNA and mRNA vaccines given that the study did not directly compare the two is appropriate.

A lingering issue is the concern raised by both reviewers about testing neutralization vs more contemporary Omicron strains (i.e. BA5). In the response to the reviewers, the authors state they analyzed neutralizing activity vs BA5 but did not detect significant activity. However, it appears this information has not been incorporated anywhere in the manuscript. "Data not shown" is not really acceptable in this case. The authors should specifically state in the paper that they did not see neutralizing responses vs BA5 and show that data in Fig 4 alongside the positive neutralizing data they showed vs other VOCs or in a supplemental figure.

Reviewer 2 :

The authors have responded fairly comprehensively to a strong set of referee reports and produced an improved manuscript.

The lack of detectable neutralisation of Omicron sub-variants is quite a strong caveat to their observations and claims - it would be sensible for them to allocate greater prominence to this point.

Response:

As the reviewers suggested, we have included neutralizing data against the live Omicron (BA.5) (52wk) to **Extended Data Figure 5a**. We have introduced the following statement in result section. (Lines 278-281 in the track change version)

“We also assessed the serum neutralizing activity against the live Omicron variant (BA.5). However, the neutralization activity against BA.5 was detected in only one animal at 52 wks post immunization (Extended Data Figure 5a).”